# LIBERO: Benchmarking Knowledge Transfer for Lifelong Robot Learning

[†]**Bo Liu**[*], [†]**Yifeng Zhu**[*], [‡]**Chongkai Gao**[*], [†]**Yihao Feng**
[†]**Qiang Liu,** [†]**Yuke Zhu,** [†,§]**Peter Stone**
[†]The University of Texas at Austin, [§]Sony AI, [‡]Tsinghua University
{bliu,yifengz,lqiang,yukez,pstone}@cs.utexas.edu
yihao.ac@gmail.com, gck20@mails.tsinghua.edu.cn

**Abstract:** Lifelong learning offers a promising paradigm of building a generalist agent that learns and adapts over its lifespan. Unlike traditional lifelong learning problems in image and text domains, which primarily involve the transfer of declarative knowledge of entities and concepts, lifelong learning in decision-making (LLDM) also necessitates the transfer of procedural knowledge, such as actions and behaviors. To advance research in LLDM, we introduce LIBERO, a novel benchmark of lifelong learning for robot manipulation. Specifically, LIBERO highlights five key research topics in LLDM: **1)** how to efficiently transfer declarative knowledge, procedural knowledge, or the mixture of both; **2)** how to design effective policy architectures and **3)** effective algorithms for LLDM; **4)** the robustness of a lifelong learner with respect to task ordering; and **5)** the effect of model pretraining for LLDM. We develop an extendible *procedural generation* pipeline that can in principle generate infinitely many tasks. For benchmarking purpose, we create four task suites (130 tasks in total) that we use to investigate the above-mentioned research topics. To support sample-efficient learning, we provide high-quality human-teleoperated demonstration data for all tasks. Our extensive experiments present several insightful or even *unexpected* discoveries: sequential finetuning outperforms existing lifelong learning methods in forward transfer, no single visual encoder architecture excels at all types of knowledge transfer, and naive supervised pretraining can hinder agents' performance in the subsequent LLDM.

## 1 Introduction

A longstanding goal in machine learning is to develop a generalist agent that can perform a wide range of tasks. While multitask learning [1] is one approach, it is computationally demanding and not adaptable to ongoing changes. Lifelong learning [2], however, offers a practical solution by amortizing the learning process over the agent's lifespan. Its goal is to leverage prior knowledge to facilitate learning new tasks (forward transfer) and use the newly acquired knowledge to enhance performance on prior tasks (backward transfer).

The main body of the lifelong learning literature has focused on how agents transfer *declarative* knowledge in visual or language tasks, which pertains to *declarative knowledge* about entities and concepts [3, 4]. Yet it is understudied how agents transfer knowledge in decision-making tasks, which involves a mixture of both *declarative* and *procedural* knowledge (knowledge about how to *do* something). Consider a scenario where a robot, initially trained to retrieve juice from a fridge, fails after learning new tasks. This could be due to forgetting the juice or fridge's location (declarative knowledge) or how to open the fridge or grasp the juice (procedural knowledge). So far, we lack methods to systematically and quantitatively analyze this complex knowledge transfer.

To bridge this research gap, this paper introduces a new simulation benchmark, LIfelong learning BEchmark on RObot manipulation tasks, LIBERO, to facilitate the systematic study of lifelong

---

[*]Equal contribution.

7th Conference on Robot Learning (CoRL 2023), Atlanta, USA.

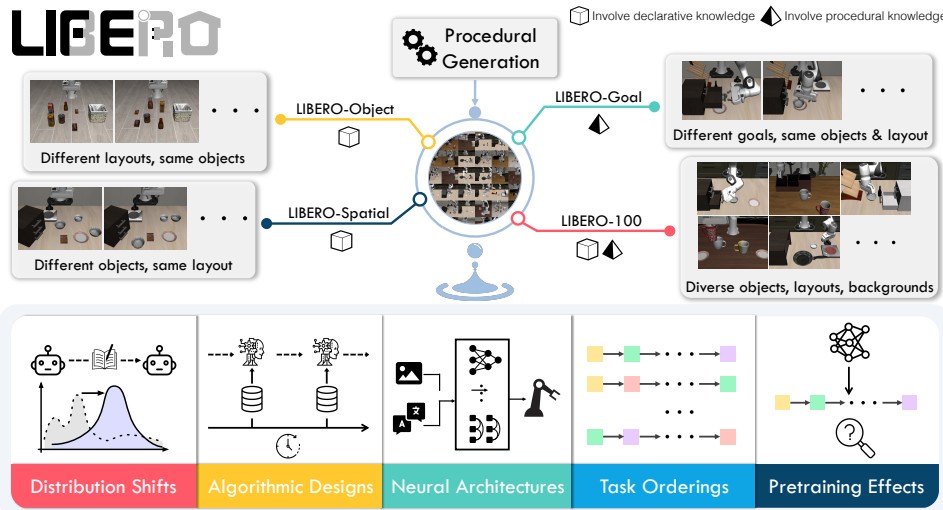

Figure 1: **Top**: LIBERO has four procedurally-generated task suites: LIBERO-SPATIAL, LIBERO-OBJECT, and LIBERO-GOAL have 10 tasks each and require transferring knowledge about spatial relationships, objects, and task goals; LIBERO-100 has 100 tasks and requires the transfer of entangled knowledge. **Bottom**: we investigate five key research topics in LLDM on LIBERO.

learning in decision making (LLDM). An ideal LLDM testbed should enable continuous learning across an expanding set of diverse tasks that share concepts and actions. LIBERO supports this through a procedural generation pipeline for endless task creation, based on robot manipulation tasks with shared visual concepts (declarative knowledge) and interactions (procedural knowledge).

For benchmarking purpose, LIBERO generates 130 language-conditioned robot manipulation tasks inspired by human activities [5] and, grouped into four suites. The four task suites are designed to examine distribution shifts in the object types, the spatial arrangement of objects, the task goals, or the mixture of the previous three (top row of Figure 1). LIBERO is scalable, extendable, and designed explicitly for studying lifelong learning in robot manipulation. To support efficient learning, we provide high-quality, human-teleoperated demonstration data for all 130 tasks.

We present an initial study using LIBERO to investigate five major research topics in LLDM (Figure 1): **1)** knowledge transfer with different types of distribution shift; **2)** neural architecture design; **3)** lifelong learning algorithm design; **4)** robustness of the learner to task ordering; and **5)** how to leverage pre-trained models in LLDM (bottom row of Figure 1). We perform extensive experiments across different policy architectures and different lifelong learning algorithms. Based on our experiments, we make several insightful or even **unexpected** observations: (1) Policy architecture design is as crucial as lifelong learning algorithms. The transformer architecture is better at abstracting temporal information than a recurrent neural network. Vision transformers work well on tasks with rich visual information (e.g., a variety of objects). Convolution networks work well when tasks primarily need procedural knowledge. (2) While the lifelong learning algorithms we evaluated are effective at preventing forgetting, they generally perform *worse* than sequential finetuning in terms of forward transfer. (3) Our experiment shows that using pretrained language embeddings of semantically-rich task descriptions yields performance *no better* than using those of the task IDs. (4) Basic supervised pretraining on a large-scale offline dataset can have a *negative* impact on the learner's downstream performance in LLDM.

## 2   Research Topics in LLDM

We outline five major research topics in LLDM that motivate the design of LIBERO and our study.

**(T1) Transfer of Different Types of Knowledge**    In order to accomplish a task such as *put the ketchup next to the plate in the basket*, a robot must understand the concept *ketchup*, the location of the *plate/basket*, and how to *put* the ketchup in the basket. Indeed, robot manipulation tasks in

general necessitate different types of knowledge, making it hard to determine the cause of failure. We present four task suites in Section 3.2: three task suites for studying the transfer of knowledge about spatial relationships, object concepts, and task goals in a disentangled manner, and one suite for studying the transfer of mixed types of knowledge.

**(T2) Neural Architecture Design**    An important research question in LLDM is how to design effective neural architectures to abstract the multi-modal observations (images, language descriptions, and robot states) and transfer only relevant knowledge when learning new tasks.

**(T3) Lifelong Learning Algorithm Design**    Given a policy architecture, it is crucial to determine what learning algorithms to apply for LLDM. Specifically, the sequential nature of LLDM suggests that even minor forgetting over successive steps can potentially lead to a total failure in execution. As such, we consider the design of lifelong learning algorithms to be an open area of research in LLDM.

**(T4) Robustness to Task Ordering**    It is well-known that task curriculum influences policy learning [6, 7]. A robot in the real world, however, often cannot choose which task to encounter first. Therefore, a good lifelong learning algorithm should be robust to different task orderings.

**(T5) Usage of Pretrained Models**    In practice, robots will be most likely pretrained on large datasets in factories before deployment [8]. However, it is not well-understood whether or how pretraining could benefit subsequent LLDM.

## 3   LIBERO

### 3.1   Procedural Generation of Tasks

Research in LLDM requires a systematic way to create new tasks while maintaining task diversity and relevance to existing tasks. LIBERO procedurally generates new tasks in three steps: **1)** extract behavioral templates from language annotations of human activities and generate sampled tasks described in natural language based on such templates; **2)** specify an initial object distribution given a task description; and **3)** specify task goals using a propositional formula that aligns with the language instructions. Our generation pipeline is built on top of `Robosuite` [9], a modular manipulation simulator that offers seamless integration. Figure 2 illustrates an example of task creation using this pipeline, and each component is expanded upon below.

**Behavioral Templates and Instruction Generation**    Human activities serve as a fertile source of tasks that can inspire and generate a vast number of manipulation tasks. We choose a large-scale activity dataset, Ego4D [5], which includes a large variety of everyday activities with language annotations. We pre-process the dataset by extracting the language descriptions and then summarize them into a large set of commonly used language templates. After this pre-processing step, we use the templates and select objects available in the simulator to generate a set of task descriptions in the form of language instructions. For example, we can generate an instruction "Open the drawer of the cabinet" from the template "Open ...".

**Initial State Distribution** ($\mu_0$)    To specify $\mu_0$, we first sample a scene layout that matches the objects/behaviors in a provided instruction. For instance, a kitchen scene is selected for an instruction *Open the top drawer of the cabinet and put the bowl in it*. Then, the details about $\mu_0$ are generated in the PDDL language [10, 11]. Concretely, $\mu_0$ contains information about object categories and their placement (Figure 2-(**A**)), and their initial status (Figure 2-(**B**)).

**Goal Specifications** ($g$)    Based on $\mu_0$ and the language instruction, we specify the task goal using a conjunction of predicates. Predicates include *unary predicates* that describe the properties of an object, such as `Open(X)` or `TurnOff(X)`, and *binary predicates* that describe spatial relations between objects, such as `On(A, B)` or `In(A, B)`. An example of the goal specification using PDDL language can be found in Figure 2-(**C**). The simulation terminates when all predicates are verified true.

### 3.2   Task Suites

While the pipeline in Section 3.1 supports the generation of an unlimited number of tasks, we offer fixed sets of tasks for benchmarking purposes. LIBERO has four task suites: LIBERO-SPATIAL, LIBERO-OBJECT, LIBERO-GOAL, and LIBERO-100. The first three task suites are curated to

disentangle the transfer of *declarative* and *procedural* knowledge (as mentioned in (T1)), while LIBERO-100 is a suite of 100 tasks with entangled knowledge transfer.

**LIBERO-X** LIBERO-SPATIAL, LIBERO-OBJECT, and LIBERO-GOAL all have 10 tasks[2] and are designed to investigate the controlled transfer of knowledge about spatial information (declarative), objects (declarative), and task goals (procedural). Specifically, all tasks in LIBERO-SPATIAL request the robot to place a bowl, among the same set of objects, on a plate. But there are two identical bowls that differ only in their location or spatial relationship to other objects. Hence, to successfully complete LIBERO-SPATIAL, the robot needs to continually learn and memorize new spatial relationships. All tasks in LIBERO-OBJECT request the robot to pick-place a unique object. Hence, to accomplish LIBERO-OBJECT, the robot needs to continually learn and memorize new object types. All tasks in LIBERO-GOAL share the same objects with fixed spatial relationships but differ only in the task goal. Hence, to accomplish LIBERO-GOAL, the robot needs to continually learn new knowledge about motions and behaviors. More details are in Appendix G.

**LIBERO-100** LIBERO-100 contains 100 tasks that entail diverse object interactions and versatile motor skills. In this paper, we split LIBERO-100 into 90 short-horizon tasks (LIBERO-90) and 10 long-horizon tasks (LIBERO-LONG). LIBERO-90 serves as the data source for pretraining **(T5)** and LIBERO-LONG for downstream evaluation of lifelong learning algorithms.

### 3.3 Lifelong Learning Algorithms

We implement three representative lifelong learning algorithms to facilitate research in algorithmic design for LLDM. Specifically, we implement Experience Replay (ER) [12], Elastic Weight Consolidation (EWC) [13], and PACKNET [14]. We pick ER, EWC, and PACKNET because they correspond to the memory-based, regularization-based, and dynamic-architecture-based methods for lifelong learning. In addition, prior research [15] has discovered that they are state-of-the-art methods. Besides these three methods, we also implement sequential finetuning (SEQL) and multitask learning (MTL), which serve as a lower bound and upper bound for lifelong learning algorithms, respectively. More details about the algorithms are in Appendix F.1.

## 4  Experiments

Experiments are conducted as an initial study for the five research topics mentioned in Section 2. Specifically, we focus on addressing the following research questions: **Q1**: How do different architectures/LL algorithms perform under specific distribution shifts? **Q2**: To what extent does neural architecture impact knowledge transfer in LLDM, and are there any discernible patterns in the specialized capabilities of each architecture? **Q3**: How do existing algorithms from lifelong supervised learning perform on LLDM tasks? **Q4**: To what extent does language embedding affect knowledge transfer in LLDM? **Q5**: How robust are different LL algorithms to task ordering in LLDM? **Q6**: Can supervised pretraining improve downstream lifelong learning performance in LLDM? The detailed results/findings are in Appendix A.

## 5  Conclusion and Limitations

This paper introduces LIBERO, a new benchmark in the robot manipulation domain for supporting research in LLDM. LIBERO includes a procedural generation pipeline that can create an infinite number of manipulation tasks in the simulator. We use this pipeline to create 130 standardized tasks and conduct a comprehensive set of experiments on policy and algorithm designs. The empirical results suggest several future research directions: 1) how to design a better neural architecture to better process spatial information or temporal information; 2) how to design a better algorithm to improve forward transfer ability; and 3) how to use pretraining to help improve lifelong learning performance. In the short term, we do not envision any negative societal impacts triggered by LIBERO. But as the lifelong learner mainly learns from humans, studying how to preserve user privacy within LLDM [16] is crucial in the long run.

---

[2]A suite of 10 tasks is enough to observe catastrophic forgetting while maintaining computation efficiency.

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

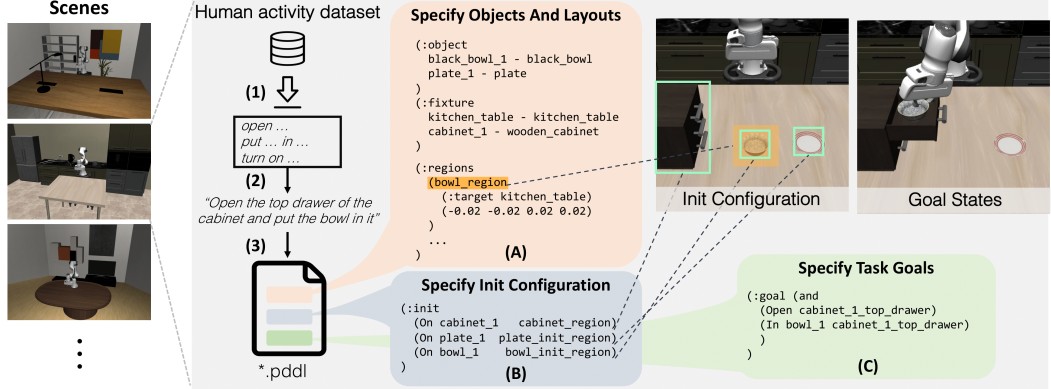

Figure 2: LIBERO's procedural generation pipeline: Extracting behavioral templates from a large-scale human activity dataset (1), Ego4D, for generating task instructions (2); Based on the task description, selecting the scene and generating the PDDL description file (3) that specifies the objects and layouts (A), the initial object configurations (B), and the task goal (C).

# A  Experiments

Experiments are conducted as an initial study for the five research topics mentioned in Section 2. We first introduce the evaluation metric used in experiments, and present analysis of empirical results in LIBERO. The detailed experimental setup is in Appendix H and the study on **Q5** is in Appendix I.2. Our experiments focus on addressing the following research questions:

**Q1**: How do different architectures/LL algorithms perform under specific distribution shifts?
**Q2**: To what extent does neural architecture impact knowledge transfer in LLDM, and are there any discernible patterns in the specialized capabilities of each architecture?
**Q3**: How do existing algorithms from lifelong supervised learning perform on LLDM tasks?
**Q4**: To what extent does language embedding affect knowledge transfer in LLDM?
**Q5**: How robust are different LL algorithms to task ordering in LLDM?
**Q6**: Can supervised pretraining improve downstream lifelong learning performance in LLDM?

## A.1  Evaluation Metrics

We report three metrics: FWT (forward transfer) [17], NBT (negative backward transfer), and AUC (area under the success rate curve). All metrics are computed in terms of success rate, as previous literature has shown that the success rate is a more reliable metric than training loss for manipulation policies [18] (Detailed explanation in Appendix I.3). Lower NBT means a policy has better performance in the previously seen tasks, higher FWT means a policy learns faster on a new task, and higher AUC means an overall better performance considering both NBT and FWT. Specifically, denote $c_{i,j,e}$ as the agent's success rate on task $j$ when it learned over $i-1$ previous tasks and has just learned $e$ epochs ($e \in \{0, 5, \ldots, 50\}$) on task $i$. Let $c_{i,i}$ be the best success rate over all evaluated epochs $e$ for the current task $i$ (i.e., $c_{i,i} = \max_e c_{i,i,e}$). Then, we find the earliest epoch $e_i^*$ in which the agent achieves the best performance on task $i$ (i.e., $e_i^* = \arg\min_e c_{i,i,e_i} = c_{i,i}$), and assume for all $e \geq e_i^*$, $c_{i,i,e} = c_{i,i}$.[3] Given a different task $j \neq i$, we define $c_{i,j} = c_{i,j,e_i^*}$. Then the three metrics are defined: $\text{FWT} = \sum_{k \in [K]} \frac{\text{FWT}_k}{K}$, $\text{FWT}_k = \frac{1}{11} \sum_{e \in \{0 \ldots 50\}} c_{k,k,e}$, $\text{NBT} = \sum_{k \in [K]} \frac{\text{NBT}_k}{K}$, $\text{NBT}_k = \frac{1}{K-k} \sum_{\tau=k+1}^{K} (c_{k,k} - c_{\tau,k})$, and $\text{AUC} = \sum_{k \in [K]} \frac{\text{AUC}_k}{K}$, $\text{AUC}_k = \frac{1}{K-k+1} (\text{FWT}_k + \sum_{\tau=k+1}^{K} c_{\tau,k})$. A visualization of these metrics is provided in Figure 4.

## A.2  Experimental Results

We present empirical results to address the research questions. Please refer to Appendix I.1 for the full results across all algorithms, policy architectures, and task suites.

---

[3]In practice, it's possible that the agent's performance on task $i$ is not monotonically increasing due to the variance of learning. But we keep the best checkpoint among those saved at epochs $\{e\}$ as if the agent stops learning after $e_i^*$.

**Study on the Policy's Neural Architectures (Q1, Q2)**    Table 1 reports the agent's lifelong learning performance using the three different neural architectures on the four task suites. Results are reported when ER and PACKNET are used as they demonstrate the best lifelong learning performance across all task suites.

| Policy Arch. | ER | | | PACKNET | | |
|---|---|---|---|---|---|---|
| | FWT(↑) | NBT(↓) | AUC(↑) | FWT(↑) | NBT(↓) | AUC(↑) |
| | | | LIBERO-LONG | | | |
| RESNET-RNN | $0.16 \pm 0.02$ | $\mathbf{0.16} \pm 0.02$ | $0.08 \pm 0.01$ | $0.13 \pm 0.00$ | $0.21 \pm 0.01$ | $0.03 \pm 0.00$ |
| RESNET-T | $\mathbf{0.48} \pm 0.02$ | $0.32 \pm 0.04$ | $\color{purple}\mathbf{0.32} \pm 0.01$ | $0.22 \pm 0.01$ | $\color{purple}\mathbf{0.08} \pm 0.01$ | $0.25 \pm 0.00$ |
| VIT-T | $0.38 \pm 0.05$ | $0.29 \pm 0.06$ | $0.25 \pm 0.02$ | $\color{purple}\mathbf{0.36} \pm 0.01$ | $0.14 \pm 0.01$ | $\color{purple}\mathbf{0.34} \pm 0.01$ |
| | | | LIBERO-SPATIAL | | | |
| RESNET-RNN | $0.40 \pm 0.02$ | $0.29 \pm 0.02$ | $0.29 \pm 0.01$ | $0.27 \pm 0.03$ | $0.38 \pm 0.03$ | $0.06 \pm 0.01$ |
| RESNET-T | $\mathbf{0.65} \pm 0.03$ | $\mathbf{0.27} \pm 0.03$ | $\mathbf{0.56} \pm 0.01$ | $0.55 \pm 0.01$ | $\mathbf{0.07} \pm 0.02$ | $\mathbf{0.63} \pm 0.00$ |
| VIT-T | $0.63 \pm 0.01$ | $0.29 \pm 0.02$ | $0.50 \pm 0.02$ | $\mathbf{0.57} \pm 0.04$ | $0.15 \pm 0.00$ | $0.59 \pm 0.03$ |
| | | | LIBERO-OBJECT | | | |
| RESNET-RNN | $0.30 \pm 0.01$ | $\mathbf{0.27} \pm 0.05$ | $0.17 \pm 0.05$ | $0.29 \pm 0.02$ | $0.35 \pm 0.02$ | $0.13 \pm 0.01$ |
| RESNET-T | $0.67 \pm 0.07$ | $0.43 \pm 0.04$ | $0.44 \pm 0.06$ | $\mathbf{0.60} \pm 0.07$ | $\mathbf{0.17} \pm 0.05$ | $\mathbf{0.60} \pm 0.05$ |
| VIT-T | $\mathbf{0.70} \pm 0.02$ | $0.28 \pm 0.01$ | $\mathbf{0.57} \pm 0.01$ | $0.58 \pm 0.03$ | $0.18 \pm 0.02$ | $0.56 \pm 0.04$ |
| | | | LIBERO-GOAL | | | |
| RESNET-RNN | $0.41 \pm 0.00$ | $0.35 \pm 0.01$ | $0.26 \pm 0.01$ | $0.32 \pm 0.03$ | $0.37 \pm 0.04$ | $0.11 \pm 0.01$ |
| RESNET-T | $\color{purple}\mathbf{0.64} \pm 0.01$ | $\mathbf{0.34} \pm 0.02$ | $\color{purple}\mathbf{0.49} \pm 0.02$ | $0.63 \pm 0.02$ | $\mathbf{0.06} \pm 0.01$ | $0.75 \pm 0.01$ |
| VIT-T | $0.57 \pm 0.00$ | $0.40 \pm 0.02$ | $0.38 \pm 0.01$ | $\mathbf{0.69} \pm 0.02$ | $0.08 \pm 0.01$ | $\mathbf{0.76} \pm 0.02$ |

Table 1: Performance of the three neural architectures using ER and PACKNET on the four task suites. Results are averaged over three seeds and we report the mean and standard error. The best performance is **bolded**, and colored in **purple** if the improvement is statistically significant over other neural architectures, when a two-tailed, Student's t-test under equal sample sizes and unequal variance is applied with a $p$-value of 0.05.

*Findings:* First, we observe that RESNET-T and VIT-T work much better than RESNET-RNN on average, indicating that using a transformer on the "temporal" level could be a better option than using an RNN model. Second, the performance difference among different architectures depends on the underlying lifelong learning algorithm. If PACKNET (a dynamic architecture approach) is used, we observe no significant performance difference between RESNET-T and VIT-T except on the LIBERO-LONG task suite where VIT-T performs much better than RESNET-T. In contrast, if ER is used, we observe that RESNET-T performs better than VIT-T on all task suites except LIBERO-OBJECT. This potentially indicates that the ViT architecture is better at processing visual information with more object varieties than the ResNet architecture when the network capacity is sufficiently large (See the MTL results in Table 8 on LIBERO-OBJECT as the supporting evidence). The above findings shed light on how one can improve architecture design for better processing of spatial and temporal information in LLDM.

**Study on Lifelong Learning Algorithms (Q1, Q3)**    Table 2 reports the lifelong learning performance of the three lifelong learning algorithms, together with the SEQL and MTL baselines. All experiments use the same RESNET-T architecture as it performs the best across all policy architectures.

*Findings:* We observed a series of interesting findings that could potentially benefit future research on algorithm design for LLDM: **1)** SEQL shows the best FWT over all task suites. This is surprising since it indicates all lifelong learning algorithms we consider actually hurt forward transfer; **2)** PACKNET outperforms other lifelong learning algorithms on LIBERO-X but is outperformed by ER significantly on LIBERO-LONG, mainly because of low forward transfer. This confirms that the dynamic architecture approach is good at preventing forgetting. But since PACKNET splits the network into different sub-networks, the essential capacity of the network for learning any individual task is smaller. Therefore, we conjecture that PACKNET is not rich enough to learn on LIBERO-

| Lifelong Algo. | FWT($\uparrow$) | NBT($\downarrow$) | AUC($\uparrow$) | FWT($\uparrow$) | NBT($\downarrow$) | AUC($\uparrow$) |
|---|---|---|---|---|---|---|
| | LIBERO-LONG | | | LIBERO-SPATIAL | | |
| SEQL | **0.54** $\pm$ 0.01 | 0.63 $\pm$ 0.01 | 0.15 $\pm$ 0.00 | **0.72** $\pm$ 0.01 | 0.81 $\pm$ 0.01 | 0.20 $\pm$ 0.01 |
| ER | 0.48 $\pm$ 0.02 | 0.32 $\pm$ 0.04 | **0.32** $\pm$ 0.01 | 0.65 $\pm$ 0.03 | 0.27 $\pm$ 0.03 | 0.56 $\pm$ 0.01 |
| EWC | 0.13 $\pm$ 0.02 | 0.22 $\pm$ 0.03 | 0.02 $\pm$ 0.00 | 0.23 $\pm$ 0.01 | 0.33 $\pm$ 0.01 | 0.06 $\pm$ 0.01 |
| PACKNET | 0.22 $\pm$ 0.01 | **0.08** $\pm$ 0.01 | 0.25 $\pm$ 0.00 | 0.55 $\pm$ 0.01 | **0.07** $\pm$ 0.02 | **0.63** $\pm$ 0.00 |
| MTL | | | 0.48 $\pm$ 0.01 | | | 0.83 $\pm$ 0.00 |
| | LIBERO-OBJECT | | | LIBERO-GOAL | | |
| SEQL | **0.78** $\pm$ 0.04 | 0.76 $\pm$ 0.04 | 0.26 $\pm$ 0.02 | **0.77** $\pm$ 0.01 | 0.82 $\pm$ 0.01 | 0.22 $\pm$ 0.00 |
| ER | 0.67 $\pm$ 0.07 | 0.43 $\pm$ 0.04 | 0.44 $\pm$ 0.06 | 0.64 $\pm$ 0.01 | 0.34 $\pm$ 0.02 | 0.49 $\pm$ 0.02 |
| EWC | 0.56 $\pm$ 0.03 | 0.69 $\pm$ 0.02 | 0.16 $\pm$ 0.02 | 0.32 $\pm$ 0.02 | 0.48 $\pm$ 0.03 | 0.06 $\pm$ 0.00 |
| PACKNET | 0.60 $\pm$ 0.07 | **0.17** $\pm$ 0.05 | **0.60** $\pm$ 0.05 | 0.63 $\pm$ 0.02 | **0.06** $\pm$ 0.01 | **0.75** $\pm$ 0.01 |
| MTL | | | 0.54 $\pm$ 0.02 | | | 0.80 $\pm$ 0.01 |

Table 2: Performance of three lifelong algorithms and the SEQL and MTL baselines on the four task suites, where the policy is fixed to be RESNET-T. Results are averaged over three seeds and we report the mean and standard error. The best performance is **bolded**, and colored in **purple** if the improvement is statistically significant over other algorithms, when a two-tailed, Student's t-test under equal sample sizes and unequal variance is applied with a $p$-value of 0.05.

LONG; **3)** EWC works worse than SEQL, showing that the regularization on the loss term can actually impede the agent's performance on LLDM problems (See Appendix I.3); and **4)** ER, the rehearsal method, is robust across all task suites.

**Study on Language Embeddings as the Task Identifier (Q4)**  To investigate to what extent language embedding play a role in LLDM, we compare the performance of the same lifelong learner using four different pretrained language embeddings. Namely, we choose BERT [19], CLIP [20], GPT-2 [21] and the Task-ID embedding. Task-ID embeddings are produced by feeding a string such as "Task 5" into a pretrained BERT model.

| Embedding Type | Dimension | FWT($\uparrow$) | NBT($\downarrow$) | AUC($\uparrow$) |
|---|---|---|---|---|
| BERT | 768 | 0.48 $\pm$ 0.02 | **0.32** $\pm$ 0.04 | 0.32 $\pm$ 0.01 |
| CLIP | 512 | **0.52** $\pm$ 0.00 | 0.34 $\pm$ 0.01 | **0.35** $\pm$ 0.01 |
| GPT-2 | 768 | 0.46 $\pm$ 0.01 | 0.34 $\pm$ 0.02 | 0.30 $\pm$ 0.01 |
| Task-ID | 768 | 0.50 $\pm$ 0.01 | 0.37 $\pm$ 0.01 | 0.33 $\pm$ 0.01 |

Table 3: Performance of a lifelong learner using four different language embeddings on LIBERO-LONG, where we fix the policy architecture to RESNET-T and the lifelong learning algorithm to ER. The Task-ID embeddings are retrieved by feeding "Task + ID" into a pretrained BERT model. Results are averaged over three seeds and we report the mean and standard error. The best performance is **bolded**. No statistically significant difference is observed among the different language embeddings.

*Findings:* From Table 3, we observe *no* statistically significant difference among various language embeddings, including the Task-ID embedding. This, we believe, is due to sentence embeddings functioning as bag-of-words that differentiates different tasks. This insight calls for better language encoding to harness the semantic information in task descriptions. Despite the similar performance, we opt for BERT embeddings as our default task embedding.

**Study on How Pretraining Affects Downstream LLDM (Q6)**  Fig 3 reports the results on LIBERO-LONG of five combinations of algorithms and policy architectures, when the underlying model is pretrained on the 90 short-horizion tasks in LIBERO-100 or learned from scratch. For pretraining, we apply behavioral cloning on the 90 tasks using the three policy architectures for 50 epochs. We save a checkpoint every 5 epochs of training and then pick the checkpoint for each architecture that has the best performance as the pretrained model for downstream LLDM.

*Findings:* We observe that the basic supervised pretraining can *hurt* the model's downstream lifelong learning performance. This, together with the results seen in Table 2 (e.g., naive sequential fine-tuning

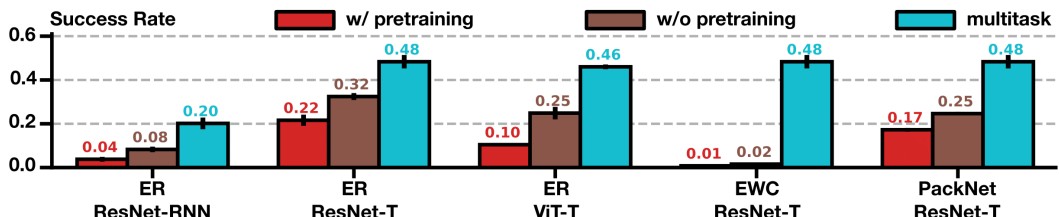

Figure 3: Performance of different combinations of algorithms and architectures without pretraining or with pretraining. The multi-task learning performance is also included for reference.

has better forward transfer than when lifelong learning algorithms are applied), indicates that better pretraining techniques are needed.

**Attention Visualization:** To better understand what type of knowledge the agent forgets during the lifelong learning process, we visualize the agent's attention map on each observed image input. The visualized saliency maps and the discussion can be found in Appendix I.4.

# B Related Work

This section provides an overview of existing benchmarks for lifelong learning and robot learning. We refer the reader to Appendix F.1 for a detailed review of lifelong learning algorithms.

**Lifelong Learning Benchmarks** Pioneering work has adapted standard vision or language datasets for studying LL. This line of work includes image classification datasets like MNIST [22], CIFAR [23], and ImageNet [24]; segmentation datasets like Core50 [25]; and natural language understanding datasets like GLUE [26] and SuperGLUE [27]. Besides supervised learning datasets, video game benchmarks (e.g., Atari [28], XLand [29], and VisDoom [30]) in reinforcement learning (RL) have also been used for studying LL. However, LL in standard supervised learning does not involve procedural knowledge transfer, while RL problems in games do not represent human activities. ContinualWorld [15] modifies the 50 manipulation tasks in MetaWorld for LL. CORA [31] builds four lifelong RL benchmarks based on Atari, Procgen [32], MiniHack [33], and ALFRED [34]. F-SIOL-310 [35] and OpenLORIS [36] are challenging real-world lifelong object learning datasets that are captured from robotic vision systems. Prior works have also analyzed different components in a LL agent [37–39], but they do not focus on robot manipulation problems.

**Robot Learning Benchmarks** A variety of robot learning benchmarks have been proposed to address challenges in meta learning (MetaWorld [40]), causality learning (CausalWorld [41]), multi-task learning [42, 43], policy generalization to unseen objects [44, 45], and compositional learning [46]. Compared to existing benchmarks in lifelong learning and robot learning, the task suites in LIBERO are curated to address the research topics of LLDM. The benchmark includes a large number of tasks based on everyday human activities that feature rich interactive behaviors with a diverse range of objects. Additionally, the tasks in LIBERO are procedurally generated, making the benchmark scalable and adaptable. Moreover, the provided high-quality human demonstration dataset in LIBERO supports and encourages learning efficiency.

# C Background

This section introduces the problem formulation and defines key terms used throughout the paper.

## C.1 Markov Decision Process for Robot Learning

A robot learning problem can be formulated as a finite-horizon Markov Decision Process: $\mathcal{M} = (\mathcal{S}, \mathcal{A}, \mathcal{T}, H, \mu_0, R)$. Here, $\mathcal{S}$ and $\mathcal{A}$ are the state and action spaces of the robot. $\mu_0$ is the initial state distribution, $R : \mathcal{S} \times \mathcal{A} \to \mathbb{R}$ is the reward function, and $\mathcal{T} : \mathcal{S} \times \mathcal{A} \to \mathcal{S}$ is the transition function. In this work, we assume a sparse-reward setting and replace $R$ with a goal predicate

$g : \mathcal{S} \rightarrow \{0, 1\}$. The robot's objective is to learn a policy $\pi$ that maximizes the expected return: $\max_\pi J(\pi) = \mathbb{E}_{s_t, a_t \sim \pi, \mu_0}[\sum_{t=1}^H g(s_t)]$.

## C.2 Lifelong Robot Learning Problem

In a *lifelong robot learning problem*, a robot sequentially learns over $K$ tasks $\{T^1, \ldots, T^K\}$ with a single policy $\pi$. We assume $\pi$ is conditioned on the task, i.e., $\pi(\cdot \mid s; T)$. For each task, $T^k \equiv (\mu_0^k, g^k)$ is defined by the initial state distribution $\mu_0^k$ and the goal predicate $g^k$.[4] We assume $\mathcal{S}, \mathcal{A}, \mathcal{T}, H$ are the same for all tasks. Up to the $k$-th task $T^k$, the robot aims to optimize

$$\max_\pi \ J_{\text{LRL}}(\pi) = \frac{1}{k} \sum_{p=1}^k \left[ \mathbb{E}_{s_t^p, a_t^p \sim \pi(\cdot; T^p), \ \mu_0^p} \left[ \sum_{t=1}^L g^p(s_t^p) \right] \right]. \tag{1}$$

An important feature of the lifelong setting is that the agent loses access to the previous $k - 1$ tasks when it learns on task $T^k$.

**Lifelong Imitation Learning** Due to the challenge of sparse-reward reinforcement learning, we consider a practical alternative setting where a user would provide a small demonstration dataset for each task in the sequence. Denote $D^k = \{\tau_i^k\}_{i=1}^N$ as $N$ demonstrations for task $T^k$. Each $\tau_i^k = (o_0, a_0, o_1, a_1, \ldots, o_{l^k})$ where $l^k \leq H$. Here, $o_t$ is the robot's sensory input, including the perceptual observation and the information about the robot's joints and gripper. In practice, the observation $o_t$ is often non-Markovian. Therefore, following works in partially observable MDPs [47], we represent $s_t$ by the aggregated history of observations, i.e. $s_t \equiv o_{\leq t} \triangleq (o_0, o_1, \ldots, o_t)$. This results in the *lifelong imitation learning problem* with the same objective as in Eq. (1). But during training, we perform behavioral cloning [48] with the following surrogate objective function:

$$\min_\pi \ J_{\text{BC}}(\pi) = \frac{1}{k} \sum_{p=1}^k \mathbb{E}_{o_t, a_t \sim D^p} \left[ \sum_{t=0}^{l^p} \mathcal{L}\big(\pi(o_{\leq t}; T^p), a_t^p\big) \right], \tag{2}$$

where $\mathcal{L}$ is a supervised learning loss, e.g., the negative log-likelihood loss, and $\pi$ is a Gaussian mixture model. Similarly, we assume $\{D^p : p < k\}$ are not fully available when learning $T^k$.

## D   Metrics Visualization

We provide a visualization of the three metrics we compute in Figure 4. For completeness, we also

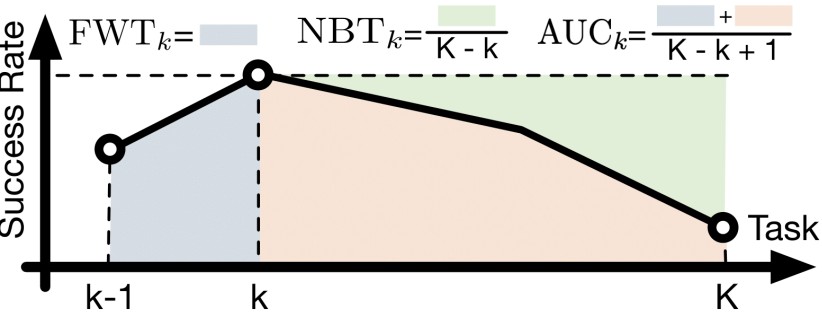

Figure 4: Metrics for LLDM

provide the formulas for the metrics here:

---

[4]Throughout the paper, a superscript/subscript is used to index the task/time step.

$$\text{FWT} = \sum_{k \in [K]} \frac{\text{FWT}_k}{K}, \quad \text{FWT}_k = \frac{1}{11} \sum_{e \in \{0...50\}} c_{k,k,e}$$

$$\text{NBT} = \sum_{k \in [K]} \frac{\text{NBT}_k}{K}, \quad \text{NBT}_k = \frac{1}{K-k} \sum_{\tau=k+1}^{K} \left( c_{k,k} - c_{\tau,k} \right)$$

$$\text{AUC} = \sum_{k \in [K]} \frac{\text{AUC}_k}{K}, \quad \text{AUC}_k = \frac{1}{K-k+1} \left( \text{FWT}_k + \sum_{\tau=k+1}^{K} c_{\tau,k} \right).$$

## E   Implemented Neural Architectures and Lifelong Learning Algorithms

| | |
|---|---|
| Neural Policy Arch. | RESNET-RNN
RESNET-T
VIT-T |
| Lifelong Learning Algo. | SEQL
EWC [13]
ER [12]
PACKNET [14]
MTL |

Table 4: The implemented neural policy architectures and the lifelong learning algorithms in LIBERO.

### E.1   Neural Network Architectures

We implement three vision-language policy networks, RESNET-RNN, RESNET-T, and VIT-T, that integrate visual, temporal, and linguistic information for LLDM. Language instructions of tasks are encoded using pretrained BERT embeddings [19]. The RESNET-RNN [18] uses a ResNet as the visual backbone that encodes per-step visual observations and an LSTM as the temporal backbone to process a sequence of encoded visual information. The language instruction is incorporated into the ResNet features using the FiLM method [49] and added to the LSTM inputs, respectively. RESNET-T architecture [50] uses a similar ResNet-based visual backbone, but a transformer decoder [51] as the temporal backbone to process outputs from ResNet, which are a temporal sequence of visual tokens. The language embedding is treated as a separate token in inputs to the transformer alongside the visual tokens. The VIT-T architecture [52], which is widely used in visual-language tasks, uses a Vision Transformer (ViT) as the visual backbone and a transformer decoder as the temporal backbone. The language embedding is treated as a separate token in inputs of both ViT and the transformer decoder. All the temporal backbones output a latent vector for every decision-making step. We compute the multi-modal distribution over manipulation actions using a Gaussian-Mixture-Model (GMM) based output head [53, 18, 54]. In the end, a robot executes a policy by sampling a continuous value for end-effector action from the output distribution. Figure 5 visualizes the three architectures.

For all the lifelong learning algorithms and neural architectures, we use behavioral cloning (BC) [48] to train policies for individual tasks (See (2)). BC allows for efficient policy learning such that we can study lifelong learning algorithms with limited computational resources. To train BC, we provide 50 trajectories of high-quality demonstrations for every single task in the generated task suites. The demonstrations are collected by human experts through teleoperation with 3Dconnexion Spacemouse.

In Section E.1, we outlined the neural network architectures utilized in our experiments, namely RESNET-RNN, RESNET-T, and VIT-T. The specifics of each architecture are illustrated in Figure 5. Furthermore, Table 5, 6, and 7 display the hyperparameters for the architectures used throughout all of our experiments.

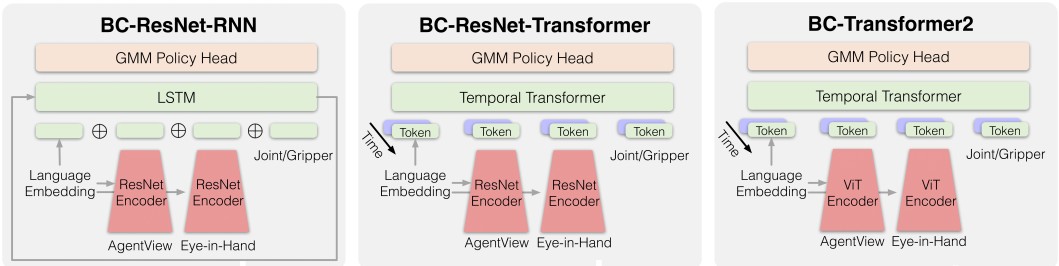

Figure 5: We provide visualizations of the architectures for RESNET-RNN, RESNET-T, and VIT-T, respectively. It is worth noting that each model architecture incorporates language embedding in distinct ways.

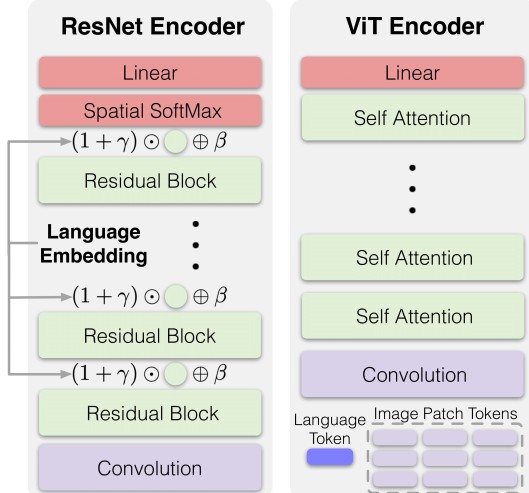

Figure 6: The image encoders: ResNet-based encoder and the vision transformer-based encoder.

## F Computation

For all experiments, we use a single Nvidia A100 GPU or a single Nvidia A40 GPU (CUDA 11.7) with 8 16 CPUs for training and evaluation.

| Variable | Value |
|---|---|
| resnet_image_embed_size | 64 |
| text_embed_size | 32 |
| rnn_hidden_size | 1024 |
| rnn_layer_num | 2 |
| rnn_dropout | 0.0 |

Table 5: Hyper parameters of RESNET-RNN.

| Variable | Value |
|---|---|
| extra_info_hidden_size | 128 |
| img_embed_size | 64 |
| transformer_num_layers | 4 |
| transformer_num_heads | 6 |
| transformer_head_output_size | 64 |
| transformer_mlp_hidden_size | 256 |
| transformer_dropout | 0.1 |
| transformer_max_seq_len | 10 |

Table 6: Hyper parameters of RESNET-T.

| Variable | Value |
|---|---|
| extra_info_hidden_size | 128 |
| img_embed_size | 128 |
| spatial_transformer_num_layers | 7 |
| spatial_transformer_num_heads | 8 |
| spatial_transformer_head_output_size | 120 |
| spatial_transformer_mlp_hidden_size | 256 |
| spatial_transformer_dropout | 0.1 |
| spatial_down_sample_embed_size | 64 |
| temporal_transformer_input_size | null |
| temporal_transformer_num_layers | 4 |
| temporal_transformer_num_heads | 6 |
| temporal_transformer_head_output_size | 64 |
| temporal_transformer_mlp_hidden_size | 256 |
| temporal_transformer_dropout | 0.1 |
| temporal_transformer_max_seq_len | 10 |

Table 7: Hyper parameters of VIT-T.

### F.1 Lifelong Learning Algorithms

Lifelong learning (LL) is a field of study that aims to understand how an agent can continually acquire and retain knowledge over an infinite sequence of tasks without catastrophically forgetting previous knowledge. Recent literature proposes three main approaches to address the problem of catastrophic forgetting in deep learning: Dynamic Architecture approaches, Regularization-Based approaches, and Rehearsal approaches. Although some recent works explore the combination of different approaches [55–57] or new strategies [58–60], our benchmark aims to provide an in-depth analysis of these three basic lifelong learning directions to reveal their pros and cons on robot learning tasks.

The dynamic architecture approach gradually expands the learning model to incorporate new knowledge [61, 62, 14, 63–65]. Regularization-based methods, on the other hand, regularize the learner to a previous checkpoint when it learns a new task [13, 66–68]. Rehearsal methods save exemplar data from prior tasks and replay them with new data to consolidate the agent's memory [12, 69–71]. For a comprehensive review of LL methods, we refer readers to surveys [72, 73].

The following paragraphs provide details on the three lifelong learning algorithms that we have implemented.

**ER** Experience Replay (ER) [12] is a **rehearsal-based** approach that maintains a memory buffer of samples from previous tasks and leverages it to learn new tasks. After the completion of policy learning for a task, ER stores a portion of the data into a storage memory. When training a new task, ER samples data from the memory and combines it with the training data from the current task so that the training data approximately represents the empirical distribution of all-task data. In our

implementation, we use a replay buffer to store a portion of the training data (up to 1000 trajectories) after training each task. For every training iteration during the training of a new task, we uniformly sample a fixed number of replay data from the memory (32 trajectories) along with each batch of training data from the new task.

**EWC**  Elastic Weight Consolidation(EWC) [13] is a **regularization-based** approach that add a regularization term that constraints neural network update to the original single-task learning objective. Specifically, EWC uses the Fisher information matrix that quantify the importance of every neural netwrk parameter. The loss function for task $k$ is:

$$\mathcal{L}_k^{EWC}(\theta) = \mathcal{L}_K^{BC}(\theta) + \sum_i \frac{\lambda}{2} F_i \left( \theta_i - \theta_{k-1,i}^* \right)^2,$$

where $\lambda$ is a penalty hyperparameter, and the coefficient $F_i$ is the diagonal of the Fisher information matrix: $F_k = \mathbb{E}_{s \sim \mathcal{D}_k} \mathbb{E}_{a \sim p_\theta(\cdot|s)} \left( \nabla_{\theta_k} \log p_{\theta_k}(a|s) \right)^2$. In this work, we use the online update version of EWC that updates the Fisher information matrix using exponential moving average along the lifelong learning process, and use the empirical estimation of above Fisher information matrix to stabilize the estimation. Formally, the actually used estimation of Fisher Information Matrix is $\tilde{F}_k = \gamma F_{k-1} + (1-\gamma) F_k$, where $F_k = \mathbb{E}_{(s,a) \sim \mathcal{D}_k} \left( \nabla_{\theta_k} \log p_{\theta_k}(a|s) \right)^2$ and $k$ is the task number. We set $\gamma = 0.9$ and $\lambda = 5 \cdot 10^4$.

**PACKNET**  PACKNET [14] is a **dynamic architecture-based** approach that aims to prevent changes to parameters that are important for previous tasks in lifelong learning. To achieve this, PACKNET iteratively trains, prunes, fine-tunes, and freezes parts of the network. The method theoretically completely avoids catastrophic forgetting, but for each new task, the number of available parameters shrinks. The pruning process in PACKNET involves two stages. First, the network is trained, and at the end of the training, a fixed proportion of the most important parameters (25% in our implementation) are chosen, and the rest are pruned. Second, the selected part of the network is fine-tuned and then frozen. In our implementation, we follow the original paper [14] and do not train all biases and normalization layers. We perform the same number of fine-tuning epochs as for training (50 epochs in our implementation). Note that all evaluation metrics are calculated *before* the fine-tuning stage.

# G   LIBERO Task Suite Designs

## G.1   Task Suites

We visualize all the tasks from the four task suites in Figure 7- 10. Figure 7 visualizes the initial states since the task goals are always the same. All the figures visualize the goal states of tasks except for Figure 7, which visualizes the initial states since the task goals are always the same.

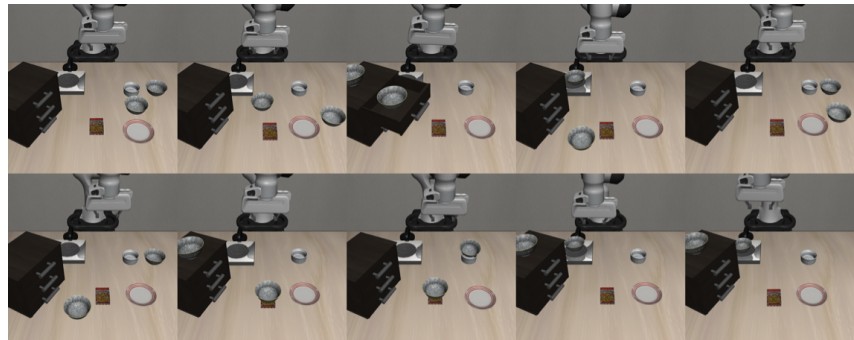

Figure 7: LIBERO-SPATIAL

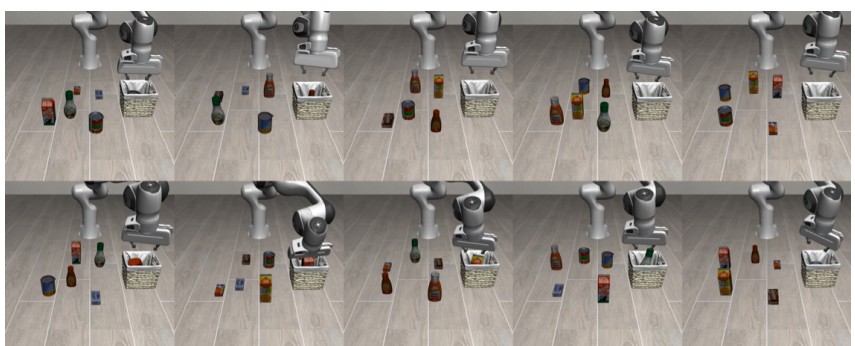

Figure 8: LIBERO-OBJECT

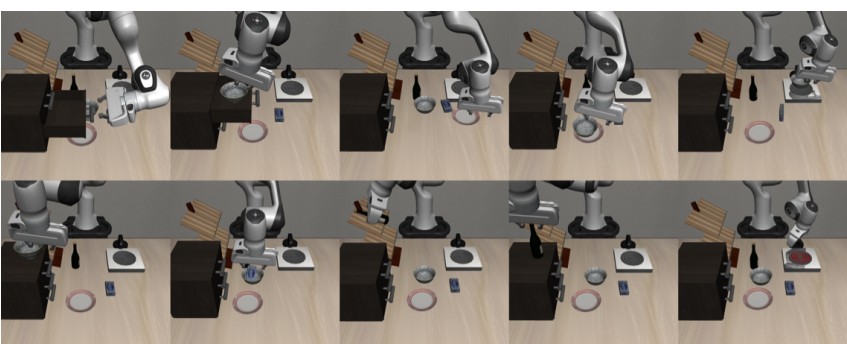

Figure 9: LIBERO-GOAL

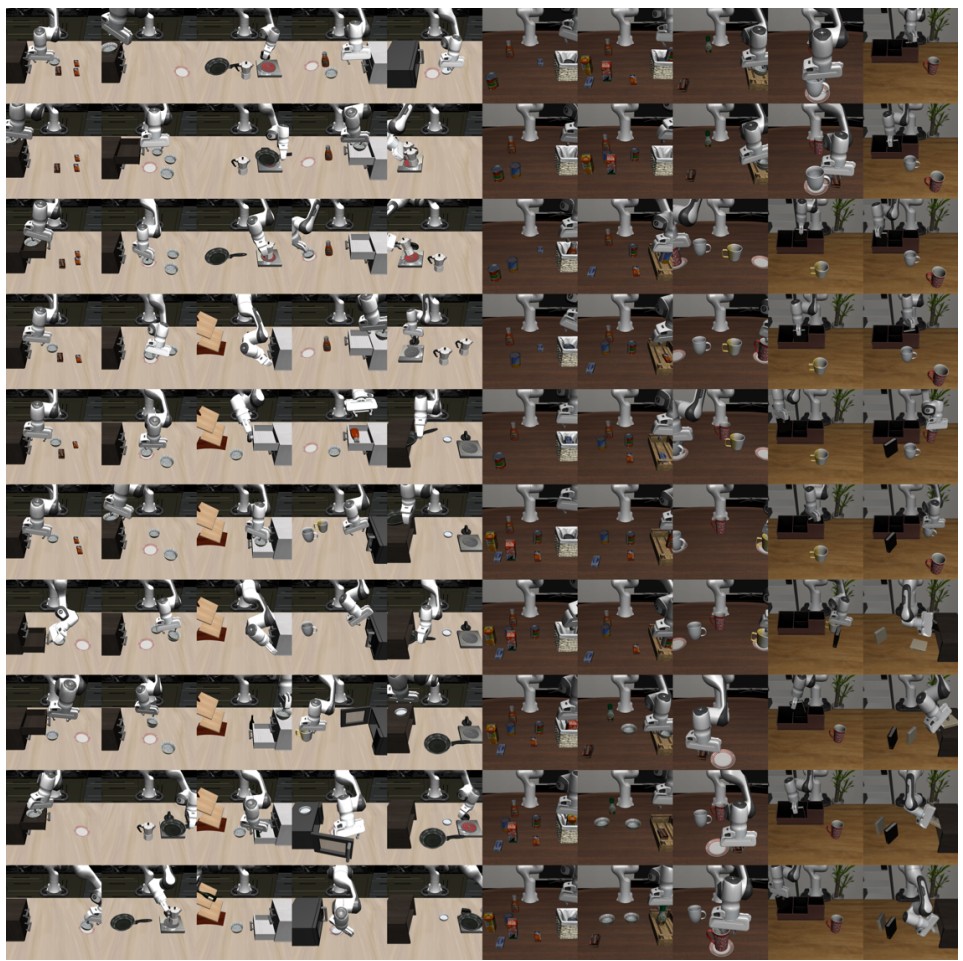

Figure 10: LIBERO-100

## G.2 PDDL-based Scene Description File

Here we visualize the whole content of an example scene description file based on PDDL. This file corresponds to the task shown in Figure 2.

**Example task:** *Open the top drawer of the cabinet and put the bowl in it*.

```
(define (problem LIBERO_Kitchen_Tabletop_Manipulation)
  (:domain robosuite)
  (:language open the top drawer of the cabinet and put the bowl in it)
    (:regions
      (wooden_cabinet_init_region
          (:target kitchen_table)
          (:ranges (
              (-0.01 -0.31 0.01 -0.29)
            )
          )
          (:yaw_rotation (
              (3.141592653589793 3.141592653589793)
            )
          )
      )
      (akita_black_bowl_init_region
          (:target kitchen_table)
          (:ranges (
              (-0.025 -0.025 0.025 0.025)
            )
          )
          (:yaw_rotation (
              (0.0 0.0)
            )
          )
      )
      (plate_init_region
          (:target kitchen_table)
          (:ranges (
              (-0.025 0.225 0.025 0.275)
            )
          )
          (:yaw_rotation (
              (0.0 0.0)
            )
          )
      )
      (top_side
          (:target wooden_cabinet_1)
      )
      (top_region
          (:target wooden_cabinet_1)
      )
      (middle_region
          (:target wooden_cabinet_1)
      )
      (bottom_region
          (:target wooden_cabinet_1)
      )
    )

  (:fixtures
```

```
        kitchen_table - kitchen_table
        wooden_cabinet_1 - wooden_cabinet
    )

    (: objects
        akita_black_bowl_1 - akita_black_bowl
        plate_1 - plate
    )

    (: obj_of_interest
        wooden_cabinet_1
        akita_black_bowl_1
    )

    (: init
        (On akita_black_bowl_1 kitchen_table_akita_black_bowl_init_region)
        (On plate_1 kitchen_table_plate_init_region)
        (On wooden_cabinet_1 kitchen_table_wooden_cabinet_init_region)
    )

    (: goal
        (And (Open wooden_cabinet_1_top_region)
             (In akita_black_bowl_1 wooden_cabinet_1_top_region)
        )
    )

)
```

# H  Experimental Setup

We consider five lifelong learning algorithms: SEQL the sequential learning baseline where the agent learns each task in the sequence directly without any further consideration, MTL the multitask learning baseline where the agent learns all tasks in the sequence simultaneously, the regularization-based method EWC [13], the replay-based method ER [12], and the dynamic architecture-based method PACKNET [14]. SEQL and MTL can be seen as approximations of the lower and upper bounds respectively for any lifelong learning algorithm. The other three methods represent the three primary categories of lifelong learning algorithms. For the neural architectures, we consider three vision-language policy architectures: RESNET-RNN, RESNET-T, VIT-T, which differ in how spatial or temporal information is aggregated (See Appendix E.1 for more details). For each task, the agent is trained over 50 epochs on the 50 demonstration trajectories. We evaluate the agent's average success rate over 20 test rollout trajectories of a maximum length of 600 every 5 epochs. We use Adam optimizer [74] with a batch size of 32, and a cosine scheduled learning rate from 0.0001 to 0.00001 for each task. Following the convention of Robomimic [18], we pick the model checkpoint that achieves the best success rate as the final policy for a given task. After 50 epochs of training, the agent with the best checkpoint is then evaluated on all previously learned tasks, with 20 test rollout trajectories for each task. All policy networks are matched in Floating Point Operations Per Second (FLOPS): all policy architectures have ∼13.5G FLOPS. For each combination of algorithm, policy architecture, and task suite, we run the lifelong learning method 3 times with random seeds {100, 200, 300} (180 experiments in total). See Table 4 for the implemented algorithms and architectures.

# I  Additional Experiment Results

## I.1  Full Results

We provide the full results across three different lifelong learning algorithms (e.g., EWC, ER, PACKNET) and three different policy architectures (e.g., RESNET-RNN, RESNET-T, VIT-T) on the four task suites in Table 8.

To better illustrate the performance of each lifelong learning agent throughout the learning process, we present plots that show how the agent's performance evolves over the stream of tasks. Firstly, we provide plots that compare the performance of the agent using different lifelong learning algorithms while fixing the policy architecture (refer to Figure 11,12, and 13). Next, we provide plots that compare the performance of the agent using different policy architectures while fixing the lifelong learning algorithm (refer to Figure14, 15, and 16)

| Algo. | Policy Arch. | FWT(↑) | NBT(↓) | AUC(↑) | FWT(↑) | NBT(↓) | AUC(↑) |
|---|---|---|---|---|---|---|---|
| | | LIBERO-LONG | | | LIBERO-SPATIAL | | |
| SEQL | RESNET-RNN | $0.24 \pm 0.02$ | $0.28 \pm 0.01$ | $0.07 \pm 0.01$ | $0.50 \pm 0.01$ | $0.61 \pm 0.01$ | $0.14 \pm 0.01$ |
| | RESNET-T | $0.54 \pm 0.01$ | $0.63 \pm 0.01$ | $0.15 \pm 0.00$ | $0.72 \pm 0.01$ | $0.81 \pm 0.01$ | $0.20 \pm 0.01$ |
| | VIT-T | $0.44 \pm 0.04$ | $0.50 \pm 0.05$ | $0.13 \pm 0.01$ | $0.63 \pm 0.02$ | $0.76 \pm 0.01$ | $0.16 \pm 0.01$ |
| ER | RESNET-RNN | $0.16 \pm 0.02$ | $0.16 \pm 0.02$ | $0.08 \pm 0.01$ | $0.40 \pm 0.02$ | $0.29 \pm 0.02$ | $0.29 \pm 0.01$ |
| | RESNET-T | $0.48 \pm 0.02$ | $0.32 \pm 0.04$ | $0.32 \pm 0.01$ | $0.65 \pm 0.03$ | $0.27 \pm 0.03$ | $0.56 \pm 0.01$ |
| | VIT-T | $0.38 \pm 0.05$ | $0.29 \pm 0.06$ | $0.25 \pm 0.02$ | $0.63 \pm 0.01$ | $0.29 \pm 0.02$ | $0.50 \pm 0.02$ |
| EWC | RESNET-RNN | $0.02 \pm 0.00$ | $0.04 \pm 0.01$ | $0.00 \pm 0.00$ | $0.14 \pm 0.02$ | $0.23 \pm 0.02$ | $0.03 \pm 0.00$ |
| | RESNET-T | $0.13 \pm 0.02$ | $0.22 \pm 0.03$ | $0.02 \pm 0.00$ | $0.23 \pm 0.01$ | $0.33 \pm 0.01$ | $0.06 \pm 0.01$ |
| | VIT-T | $0.05 \pm 0.02$ | $0.09 \pm 0.03$ | $0.01 \pm 0.00$ | $0.32 \pm 0.03$ | $0.48 \pm 0.03$ | $0.06 \pm 0.01$ |
| PACKNET | RESNET-RNN | $0.13 \pm 0.00$ | $0.21 \pm 0.01$ | $0.03 \pm 0.00$ | $0.27 \pm 0.03$ | $0.38 \pm 0.03$ | $0.06 \pm 0.01$ |
| | RESNET-T | $0.22 \pm 0.01$ | $0.08 \pm 0.01$ | $0.25 \pm 0.00$ | $0.55 \pm 0.01$ | $0.07 \pm 0.02$ | $0.63 \pm 0.00$ |
| | VIT-T | $0.36 \pm 0.01$ | $0.14 \pm 0.01$ | $0.34 \pm 0.01$ | $0.57 \pm 0.04$ | $0.15 \pm 0.00$ | $0.59 \pm 0.03$ |
| MTL | RESNET-RNN | | | $0.20 \pm 0.01$ | | | $0.61 \pm 0.00$ |
| | RESNET-T | | | $0.48 \pm 0.01$ | | | $0.83 \pm 0.00$ |
| | VIT-T | | | $0.46 \pm 0.00$ | | | $0.79 \pm 0.01$ |
| | | LIBERO-OBJECT | | | LIBERO-GOAL | | |
| SEQL | RESNET-RNN | $0.48 \pm 0.03$ | $0.53 \pm 0.04$ | $0.15 \pm 0.01$ | $0.61 \pm 0.01$ | $0.73 \pm 0.01$ | $0.16 \pm 0.00$ |
| | RESNET-T | $0.78 \pm 0.04$ | $0.76 \pm 0.04$ | $0.26 \pm 0.02$ | $0.77 \pm 0.01$ | $0.82 \pm 0.01$ | $0.22 \pm 0.00$ |
| | VIT-T | $0.76 \pm 0.03$ | $0.73 \pm 0.03$ | $0.27 \pm 0.02$ | $0.75 \pm 0.01$ | $0.85 \pm 0.01$ | $0.20 \pm 0.01$ |
| ER | RESNET-RNN | $0.30 \pm 0.01$ | $0.27 \pm 0.05$ | $0.17 \pm 0.05$ | $0.41 \pm 0.00$ | $0.35 \pm 0.01$ | $0.26 \pm 0.01$ |
| | RESNET-T | $0.67 \pm 0.07$ | $0.43 \pm 0.04$ | $0.44 \pm 0.06$ | $0.64 \pm 0.01$ | $0.34 \pm 0.02$ | $0.49 \pm 0.02$ |
| | VIT-T | $0.70 \pm 0.02$ | $0.28 \pm 0.01$ | $0.57 \pm 0.01$ | $0.57 \pm 0.00$ | $0.40 \pm 0.02$ | $0.38 \pm 0.01$ |
| EWC | RESNET-RNN | $0.17 \pm 0.04$ | $0.23 \pm 0.04$ | $0.06 \pm 0.01$ | $0.16 \pm 0.01$ | $0.22 \pm 0.01$ | $0.06 \pm 0.01$ |
| | RESNET-T | $0.56 \pm 0.03$ | $0.69 \pm 0.02$ | $0.16 \pm 0.02$ | $0.32 \pm 0.02$ | $0.48 \pm 0.03$ | $0.06 \pm 0.00$ |
| | VIT-T | $0.57 \pm 0.03$ | $0.64 \pm 0.03$ | $0.23 \pm 0.00$ | $0.32 \pm 0.04$ | $0.45 \pm 0.04$ | $0.07 \pm 0.01$ |
| PACKNET | RESNET-RNN | $0.29 \pm 0.02$ | $0.35 \pm 0.02$ | $0.13 \pm 0.01$ | $0.32 \pm 0.03$ | $0.37 \pm 0.04$ | $0.11 \pm 0.01$ |
| | RESNET-T | $0.60 \pm 0.07$ | $0.17 \pm 0.05$ | $0.60 \pm 0.05$ | $0.63 \pm 0.02$ | $0.06 \pm 0.01$ | $0.75 \pm 0.01$ |
| | VIT-T | $0.58 \pm 0.03$ | $0.18 \pm 0.02$ | $0.56 \pm 0.04$ | $0.69 \pm 0.02$ | $0.08 \pm 0.01$ | $0.76 \pm 0.02$ |
| MTL | RESNET-RNN | | | $0.10 \pm 0.03$ | | | $0.59 \pm 0.00$ |
| | RESNET-T | | | $0.54 \pm 0.02$ | | | $0.80 \pm 0.01$ |
| | VIT-T | | | $0.78 \pm 0.02$ | | | $0.82 \pm 0.01$ |

Table 8: We present the full results of all networks and algorithms on all four task suites. For each task suite, we highlight the top three AUC scores among the combinations of the three lifelong learning algorithms and the three neural architectures. The best three results are highlighted in **magenta** (the best), **light magenta** (the second best), and super light magenta (the third best), respectively.

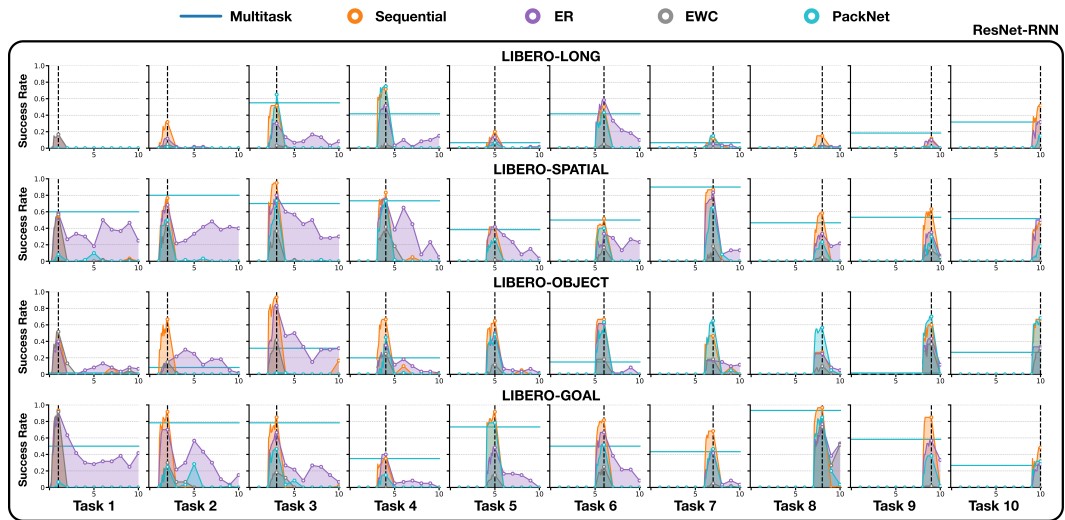

Figure 11: We compare the performance of different algorithms using the RESNET-RNN policy architecture in Figure 11. The $y$-axis represents the success rate, and the $x$-axis shows the agent's performance on each of the 10 tasks in a specific task suite over the course of learning. For example, the upper-left plot in the figure displays the agent's performance on the first task as it learns the 10 tasks sequentially.

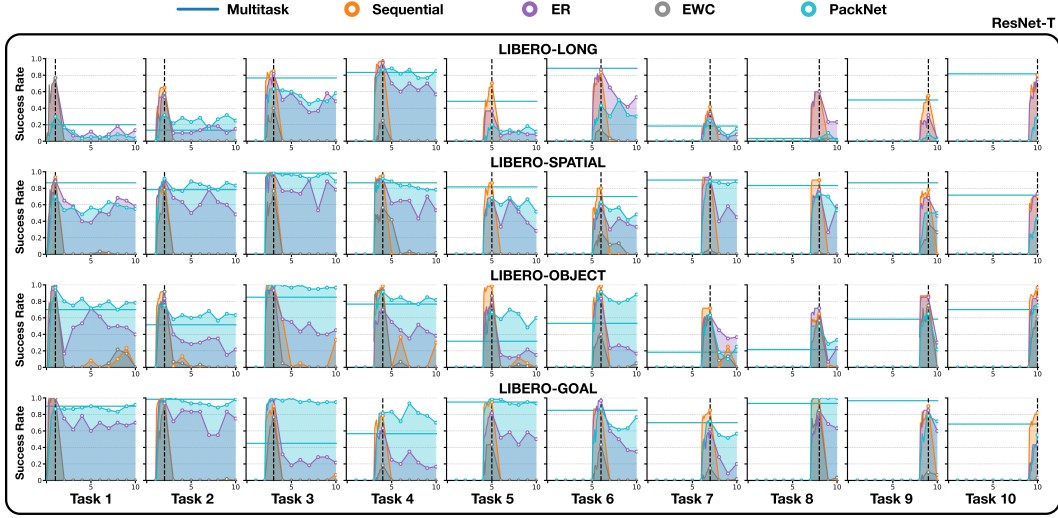

Figure 12: Comparison of different algorithms using the RESNET-T policy architecture. The $y$-axis represents the success rate, while the $x$-axis shows the agent's performance on each of the 10 tasks in a given task suite during the course of learning. For example, the plot in the upper-left corner depicts the agent's performance on the first task as it learns the 10 tasks sequentially.

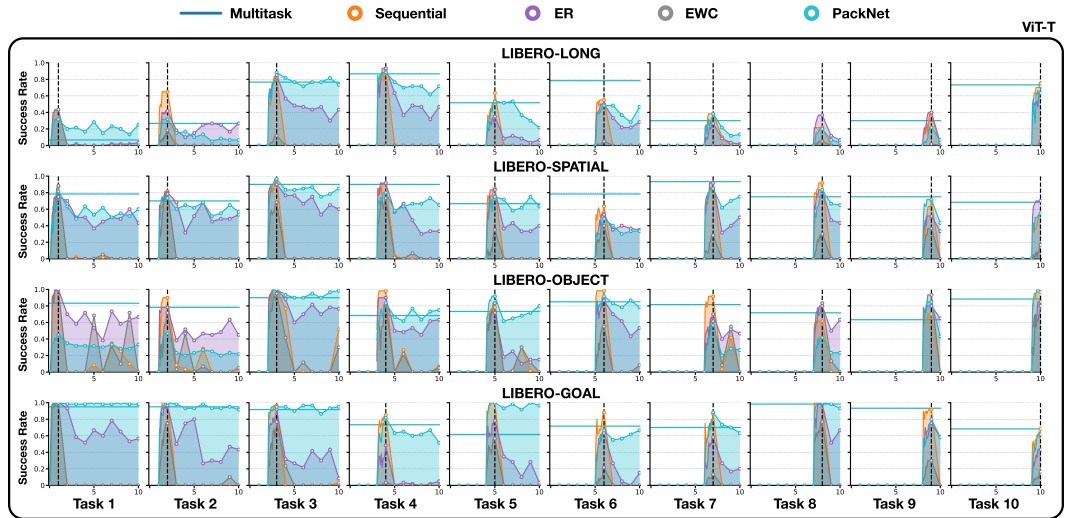

Figure 13: Comparison of different algorithms using the VIT-T policy architecture. The success rate is represented on the $y$-axis, while the $x$-axis shows the agent's performance on the 10 tasks in a given task suite over the course of learning. For instance, the plot in the upper-left corner illustrates the agent's performance on the first task when learning the 10 tasks sequentially.

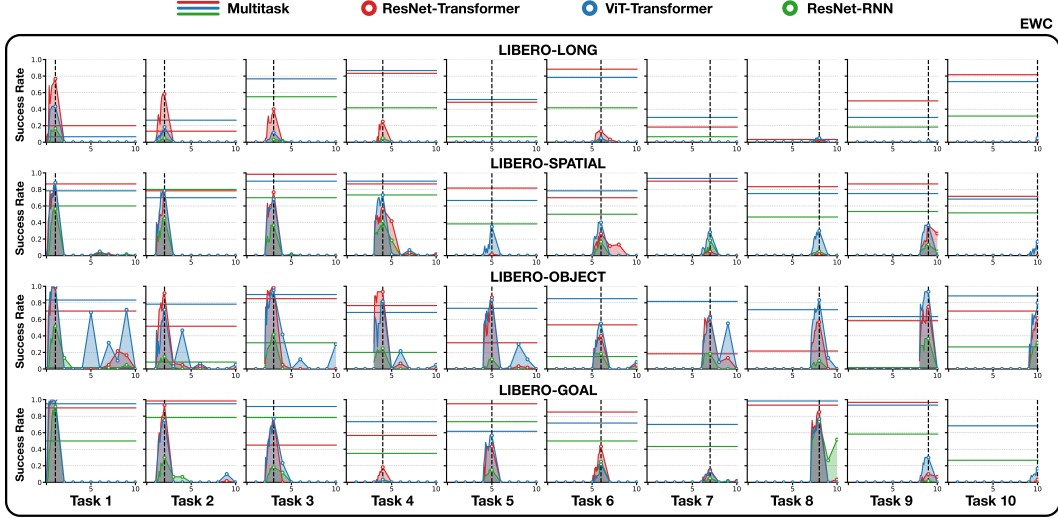

Figure 14: Comparison of different architectures with the EWC algorithm. The $y$-axis is the success rate, while the $x$-axis shows the agent's performance on the 10 tasks in a given task suite over the course of learning. For instance, the upper-left plot shows the agent's performance on the first task when learning the 10 tasks sequentially.

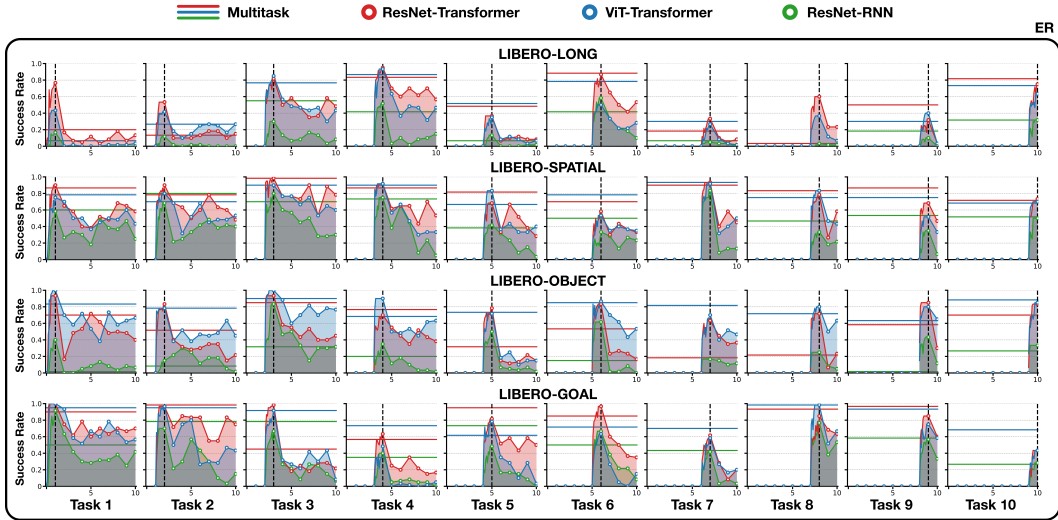

Figure 15: Comparison of different architectures with the ER algorithm. The $y$-axis is the success rate, while the $x$-axis shows the agent's performance on the 10 tasks in a given task suite ver the course of learning. For instance, the upper-left plot shows the agent's performance on the first task when learning the 10 tasks sequentially.

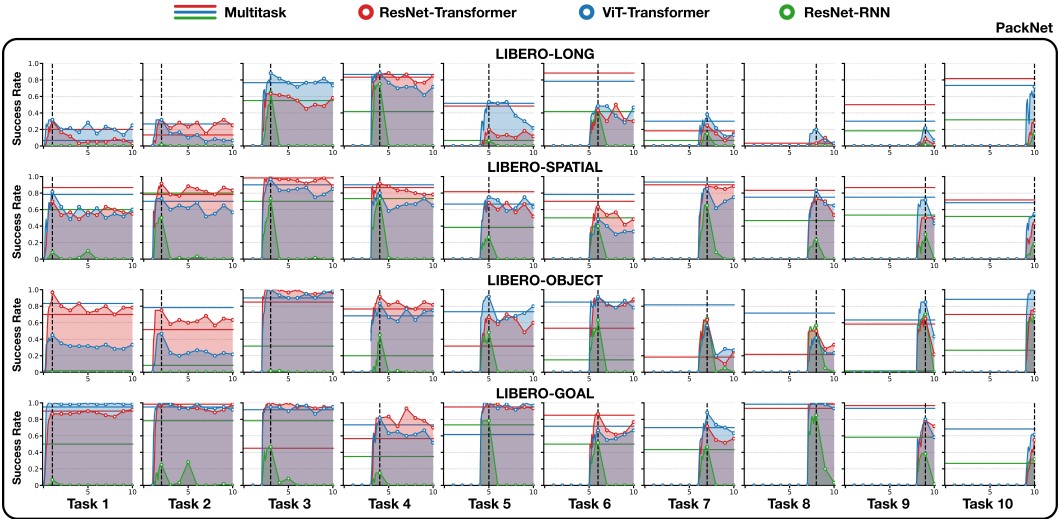

Figure 16: Comparison of different architectures with the PACKNET algorithm. The $y$-axis is the success rate, while the $x$-axis shows the agent's performance on the 10 tasks in a given task suite over the course of learning. For instance, the upper-left plot shows the agent's performance on the first task when learning the 10 tasks sequentially.

## I.2 Study on task ordering (Q4)

Figure 17 shows the result of the study on **Q4**. For all experiments in this study, we used RESNET-T as the neural architecture and evaluated both ER and PACKNET. As the figure illustrates, the performance of both algorithms varies across different task orderings. This finding highlights an important direction for future research: developing algorithms or architectures that are robust to varying task orderings.

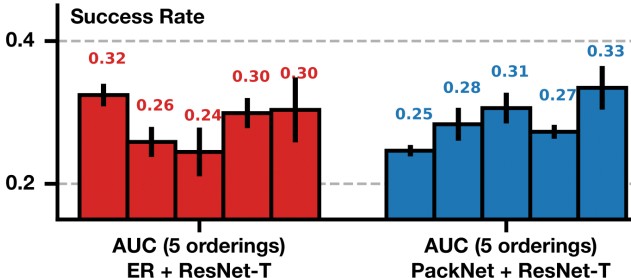

Figure 17: Performance of ER and PACKNET using RESNET-T on five different task orderings. An error bar shows the performance standard deviation for a fixed ordering.

*Findings:* From Figure 17, we observe that indeed different task ordering could result in very different performances for the same algorithm. Specifically, such difference is statistically significant for PACKNET.

## I.3 Loss v.s. Success Rates

We demonstrate that behavioral cloning loss can be a misleading indicator of task success rate in this section. In supervised learning tasks like image classifications, lower loss often indicates better prediction accuracy. However, this is not, in general, true for decision-making tasks. This is because errors can compound until failures during executing a robot [75]. Figure 18, 12 and 13 plots the training loss and success rates of three lifelong learning methods (ER, EWC, and PACKNET) for comparison. We evaluate the three algorithms on four task suites using three different neural architectures.

*Findings:* We observe that though sometimes EWC has the **lowest** loss, it did not achieve good success rate. ER, on the other hand, can have the highest loss but perform better than EWC. In conclusion, success rates, instead of behavioral cloning loss, should be the right metric to evaluate whether a model checkpoint is good or not.

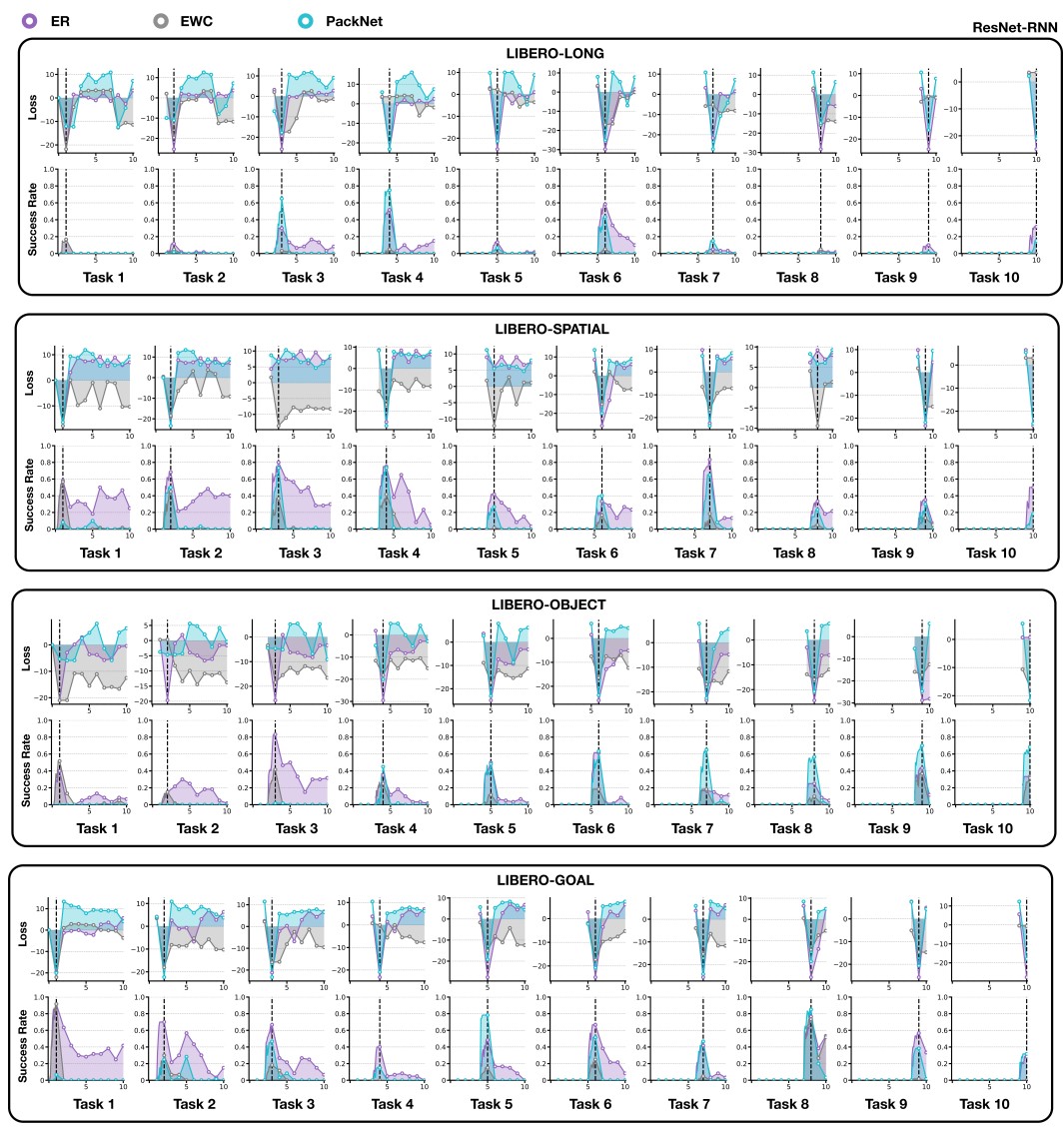

Figure 18: Losses and success rates of ER (violet), EWC (grey), and PACKNET (blue) on four task suites with RESNET-RNN policy. The first (second) row shows the loss (success rate) of the agent on task $i$ throughout the LLDM procedure.

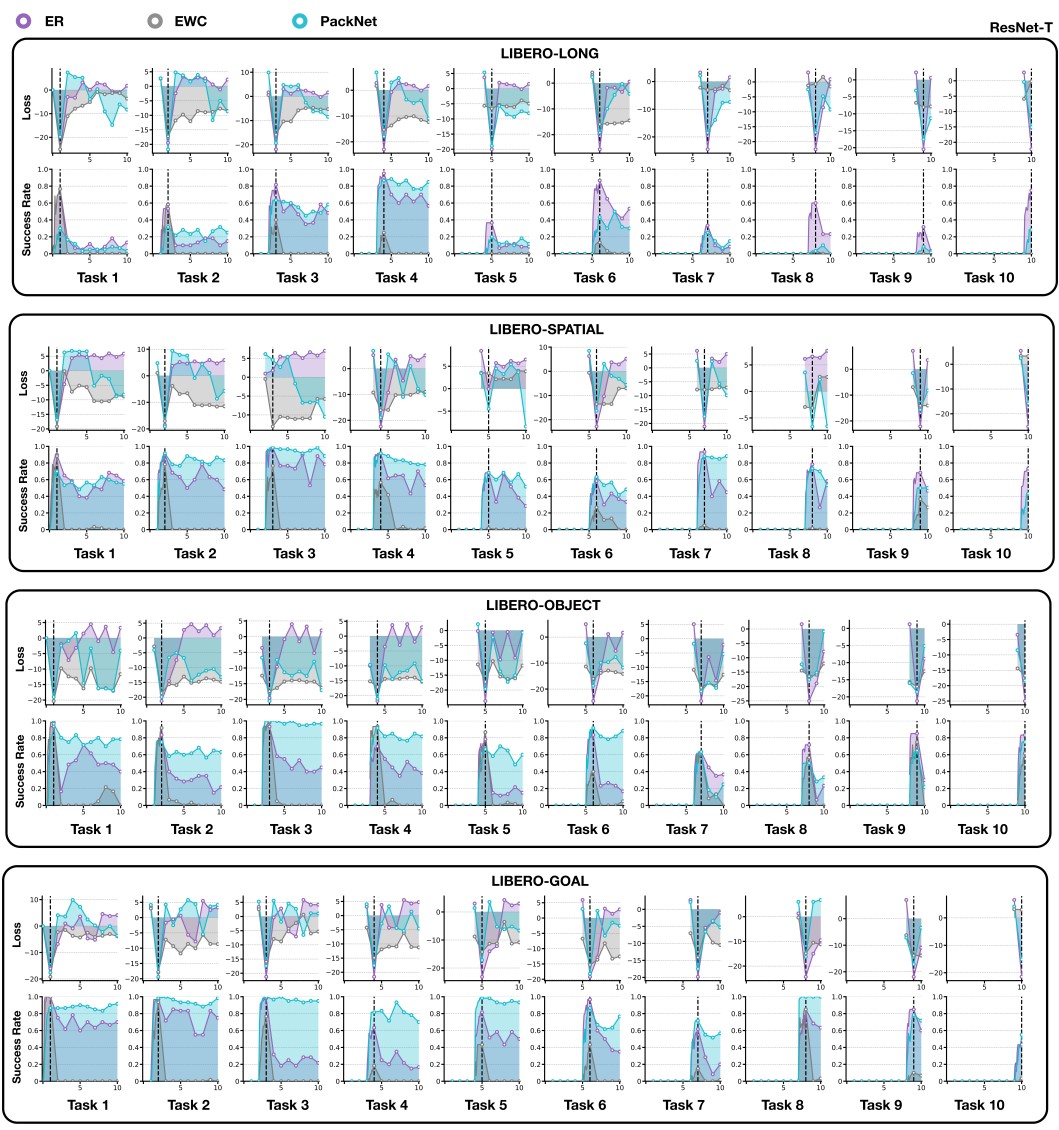

Figure 19: Losses and success rates of ER (violet), EWC (grey), and PACKNET (blue) on four task suites with RESNET-T policy. The first (second) row shows the loss (success rate) of the agent on task $i$ throughout the LLDM procedure.

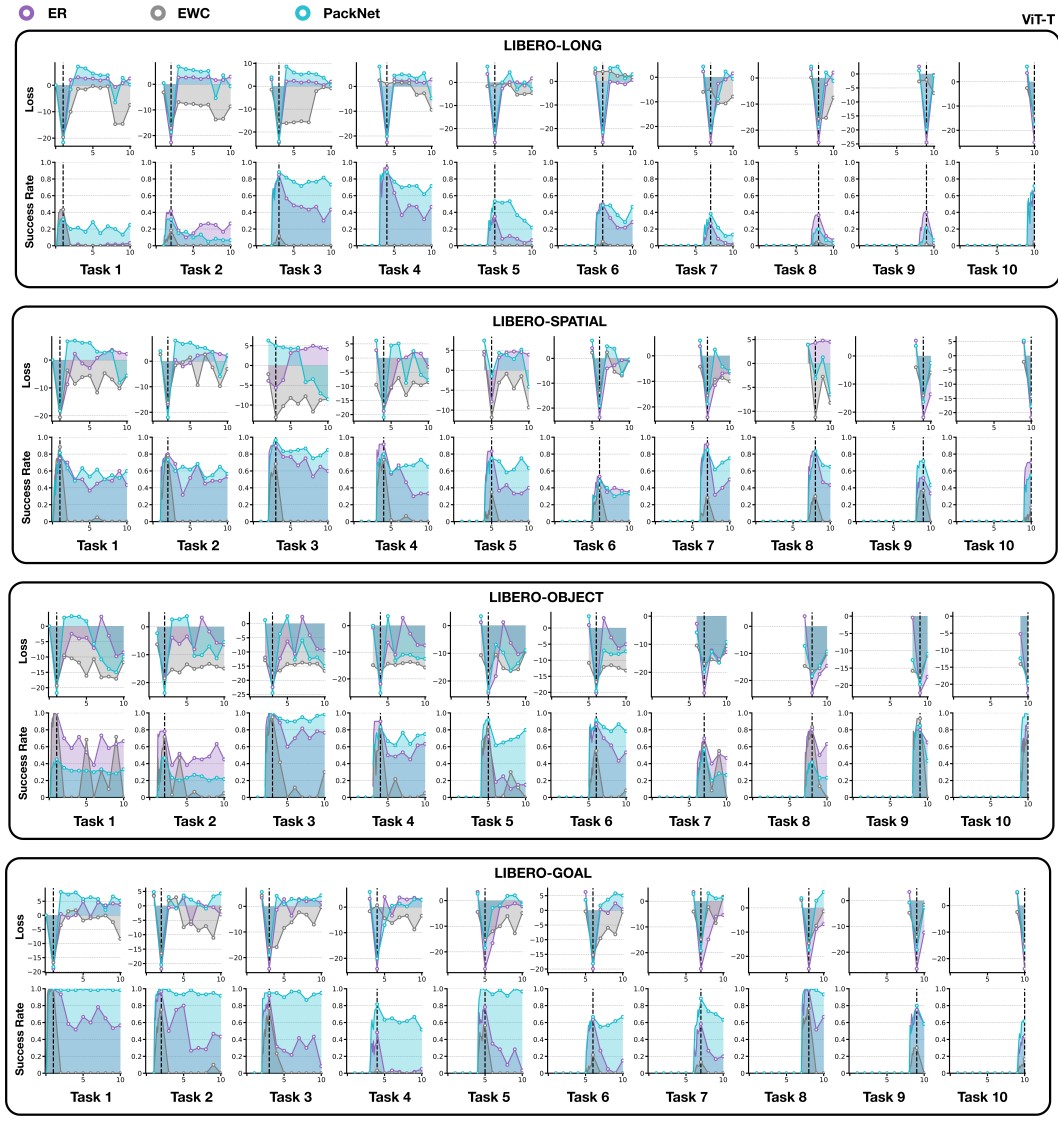

Figure 20: Losses and success rates of ER (violet), EWC (grey), and PACKNET (blue) on four task suites with VIT-T policy. The first (second) row shows the loss (success rate) of the agent on task $i$ throughout the LLDM procedure.

### I.4 Attention Visualization

It is also important to visualize the behavior of the robot and its attention maps during the completion of tasks in the lifelong learning process to give us intuition and qualitative feedback on the performance of different algorithms and architectures. We visualize the attention maps of learned policies with Greydanus et al. [76] and compare them in different studies as in A.2 to see if the robot correctly pays attention to the right regions of interest in each task.

**Perturbation-based attention visualization:** We use a perturbation-based method [76] to extract attention maps from agents. Given an input image $I$, the method applies a Gaussian filter to a pixel location $(i, j)$ to blur the image partially, and produces the perturbed image $\Phi(I, i, j)$. Denote the learned policy as $\pi$ and the inputs to the spatial module (e.g., the last latent representation of resnet or ViT encoder) $\pi_u(I)$ for image $I$. Then we define the saliency score as the Euclidean distance between the latent representations of the original and the blurred images:

$$S_\pi(i, j) = \frac{1}{2} \left\| \pi_u(I) - \pi_u(\Phi(I, i, j)) \right\|^2. \tag{3}$$

Intuitively, $S_\pi(i, j)$ describes *how much removing information from the region around location $(i, j)$ changes the policy*. In other words, a large $S_\pi(i, j)$ indicates that the information around pixel $(i, j)$ is important for the learning agent's decision-making. Instead of calculating the score for every pixel, [76] found that computing a saliency score for pixel $i$ mod 5 and $j$ mod 5 produced good saliency maps at lower computational costs for Atari games. The final saliency map $P$ is normalized as $P(i, j) = \frac{S_\pi(i,j)}{\sum_{i,j} S_\pi(i,j)}$.

We provide the visualization and our analysis on the following pages.

## Different Task Suites

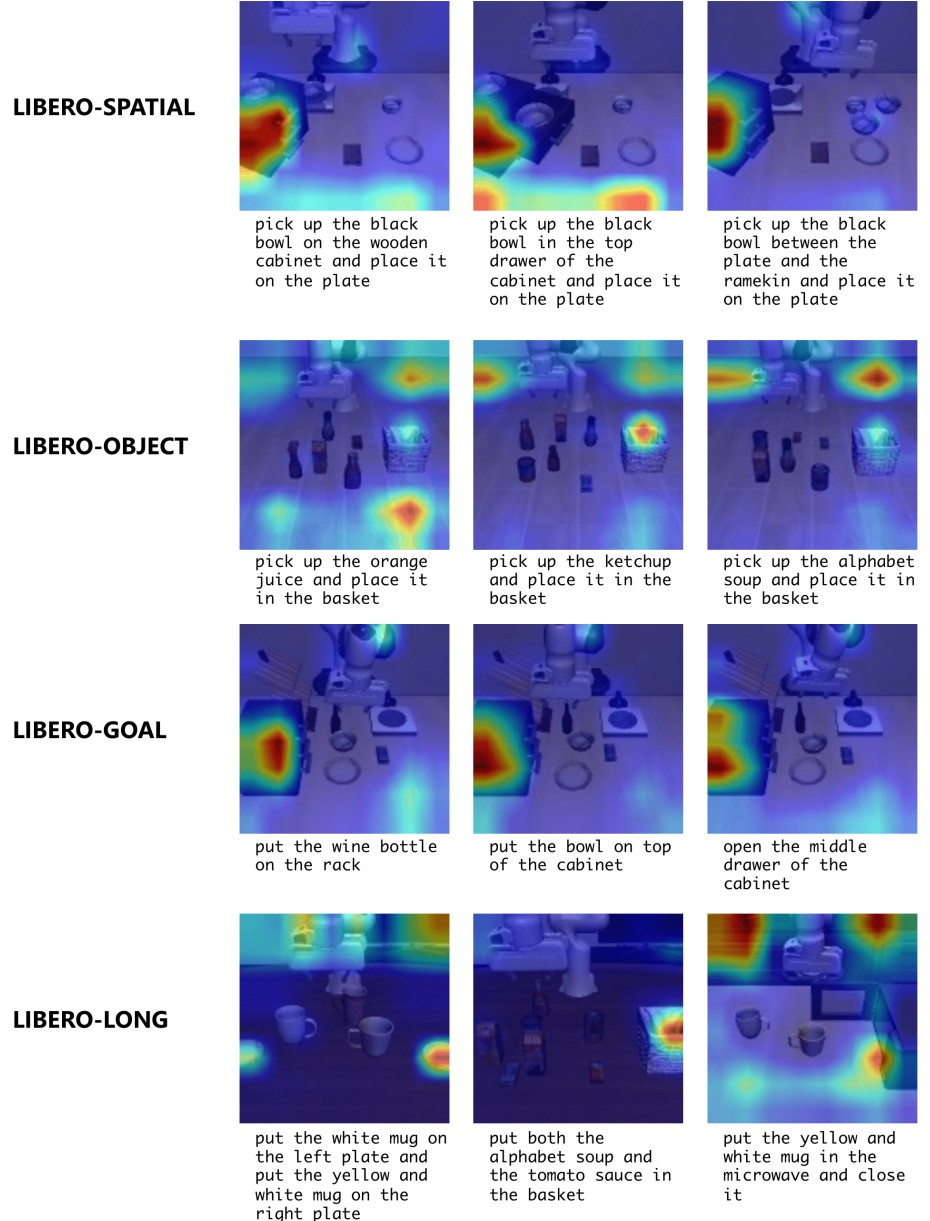

Figure 21: Attention map comparison among different task suites with ER and RESNET-T. Each row corresponds to a task suite.

*Findings:* Figure 21 shows attention visualization for 12 tasks across 4 task suites (e.g., 3 tasks per suite). We observe that:

1. policies pay more attention to the robot arm and the target placement area than the target object.

2. sometimes the policy pays attention to task-irrelevant areas, such as the blank area on the table.

These observations demonstrate that the learned policy use perceptual data for decision-making in a very different way from how humans do. The robot policies tends to spuriously correlate task-irrelevant features with actions, a major reason why the policies overfit to the tasks and do not generalize well across tasks.

**The Same Task over the Course of Lifelong Learning**

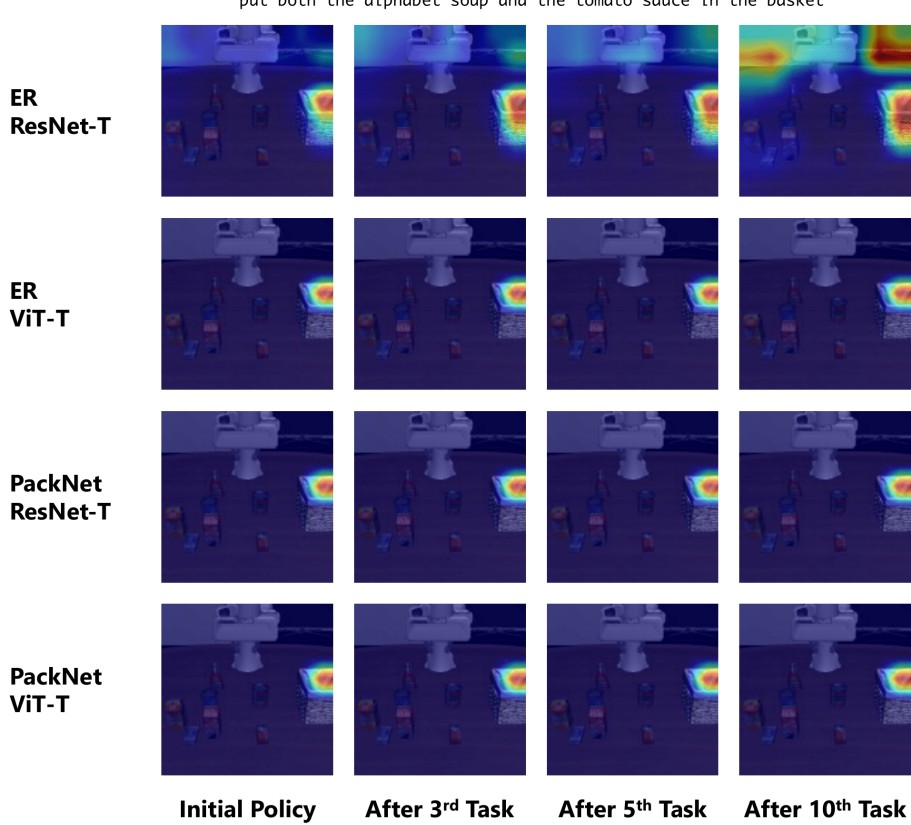

Figure 22: Attention map of the same state of the task *put both the alphabet soup and the tomato sauce in the basket* from LIBERO-LONG during lifelong learning. Each row visualizes how the attention maps change on the first task with one of the LL algorithms (ER and PACKNET) and one of the neural architectures (RESNET-T and VIT-T). Initial policy is the policy that is trained on the first task. And all the following attention maps correspond to policies after training on the third, fifth, and the tenth tasks.

*Findings:* Figure 22 shows attention visualizations from policies trained with ER and PACKNET using the architectures RESNET-T and VIT-T respectively. We observe that:

1. The ViT visual encoder's attention is more consistent over time, while the ResNet encoder's attention map gradually dilutes.

2. PackNet, as it splits the model capacity for different tasks, shows a more consistent attention map over the course of learning.

**Different Lifelong Learning Algorithms**

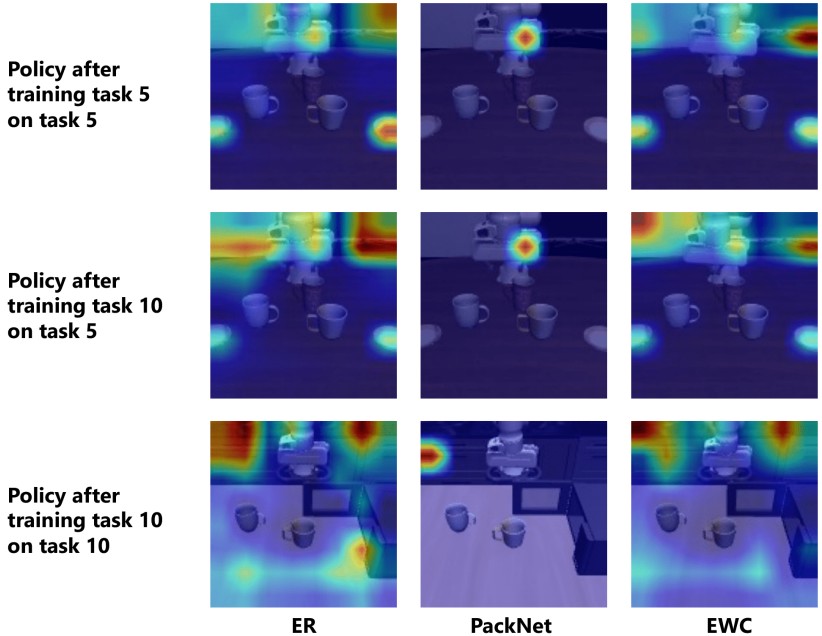

Figure 23: Comparison of attention maps of different lifelong learning algorithms with RESNET-T on LIBERO-LONG. Each row shows the same state of a task with different neural architectures. "Task 5" refers to the task *put the white mug on the left plate and put the yellow and white mug on the right plate*. "Task 10" refers to the task *put the yellow and white mug in the microwave and close it*. The second row shows the policy that is trained on task 10 and gets evaluated on task 5, showing the attention map differences in backward transfer.

*Findings:* Figure 23 shows the attention visualization of three lifelong learning algorithms on LIBERO-LONG with RESNET-T on two tasks (task 5 and task 10). The first and third rows show the attention of the policy on the same task it has just learned. While the second row shows the attention of the policy on the task it learned in the past. We observe that:

1. PACKNET shows more concentrated attention compared against ER and EWC (usually just a single mode).
2. ER shares similar attention map with EWC, but EWC performs much worse than ER. Therefore, attention can only assist the analysis but cannot be treated as a criterion for performance prediction.

**Different Neural Architectures**

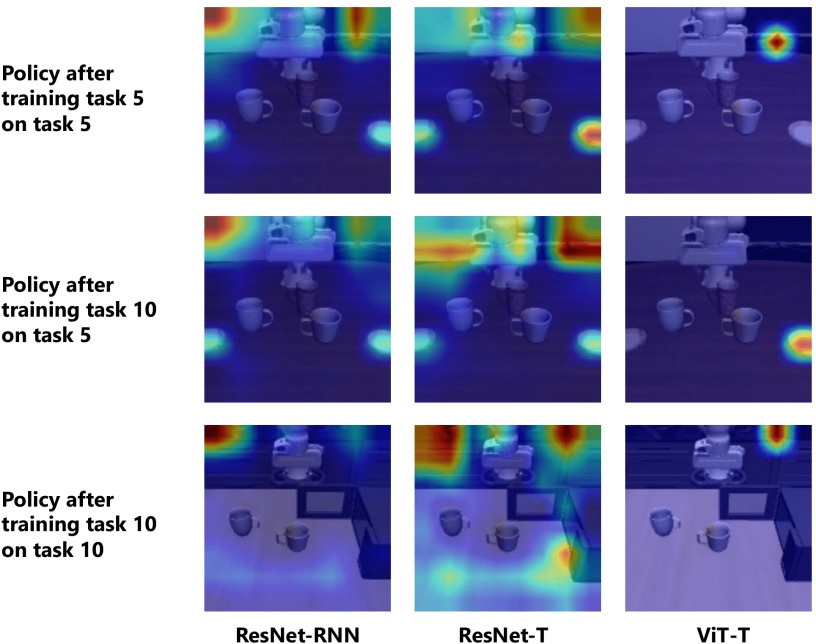

Figure 24: Comparison of attention maps of different neural architectures with ER on LIBERO-LONG. Each row shows the same state of a task with different neural architectures. "Task 5" refers to the task *put the white mug on the left plate and put the yellow and white mug on the right plate*. "Task 10" refers to the task *put the yellow and white mug in the microwave and close it*. The second row shows the policy that is trained on task 10 and gets evaluated on task 5, showing the attention map differences in backward transfer.

*Findings:* Figure 24 shows attention map comparisons of the three neural architectures on LIBERO-LONG with ER on two tasks (task 5 and task 10). We observe that:

1. ViT has more concentrated attention than policies using ResNet.

2. When ResNet forgets, the attention is changing smoothly (more diluted). But for ViT, when it forgets, the attention can completely shift to a different location.

3. When ResNet is combined with LSTM or a temporal transformer, the attention hints at the "course of future trajectory". But we do not observe that when ViT is used as the encoder.

**Different Task Ordering**

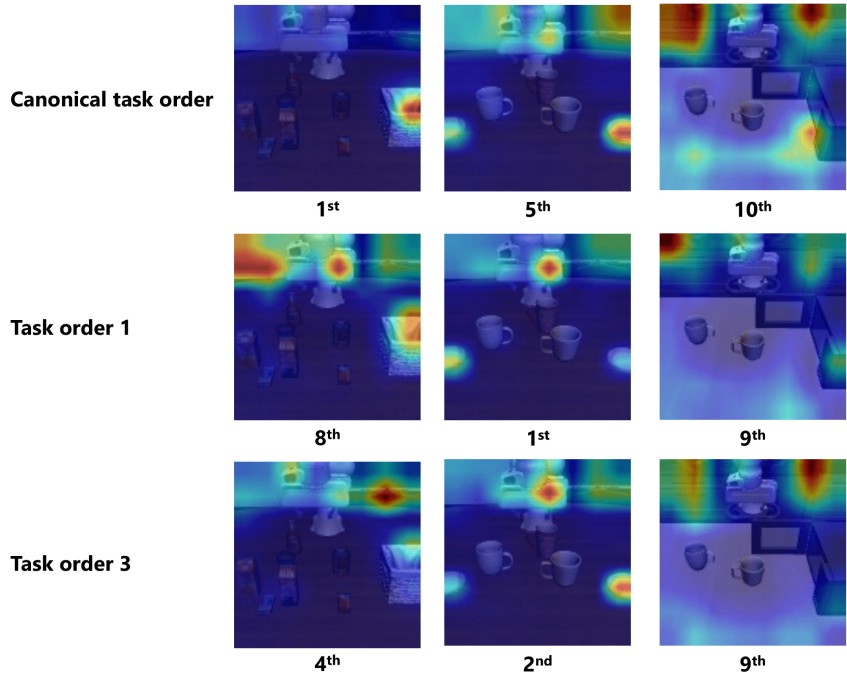

Figure 25: Attention map comparison among different orderings with ER and RESNET-T on three selected tasks from LIBERO-LONG: *put both the alphabet soup and the tomato sauce in the basket*, *put the white mug on the left plate and put the yellow and white mug on the right plate*, and *put the yellow and white mug in the microwave and close it*. Each row corresponds to a specific sequence of task ordering, and the caption of each attention map indicates the order of the task in that sequence.

*Findings:* Figure 25 shows attention map comparisons of three different task orderings. We show two immediately learned tasks from LIBERO-LONG trained with ER and RESNET-T. We observe that:

1. As expected, learning the same task at different positions in the task stream results in different attention visualization.

2. There seems to be a trend that the policy has a more spread-out attention when it learns on tasks that are later in the sequence.

**With or Without Pretraining**

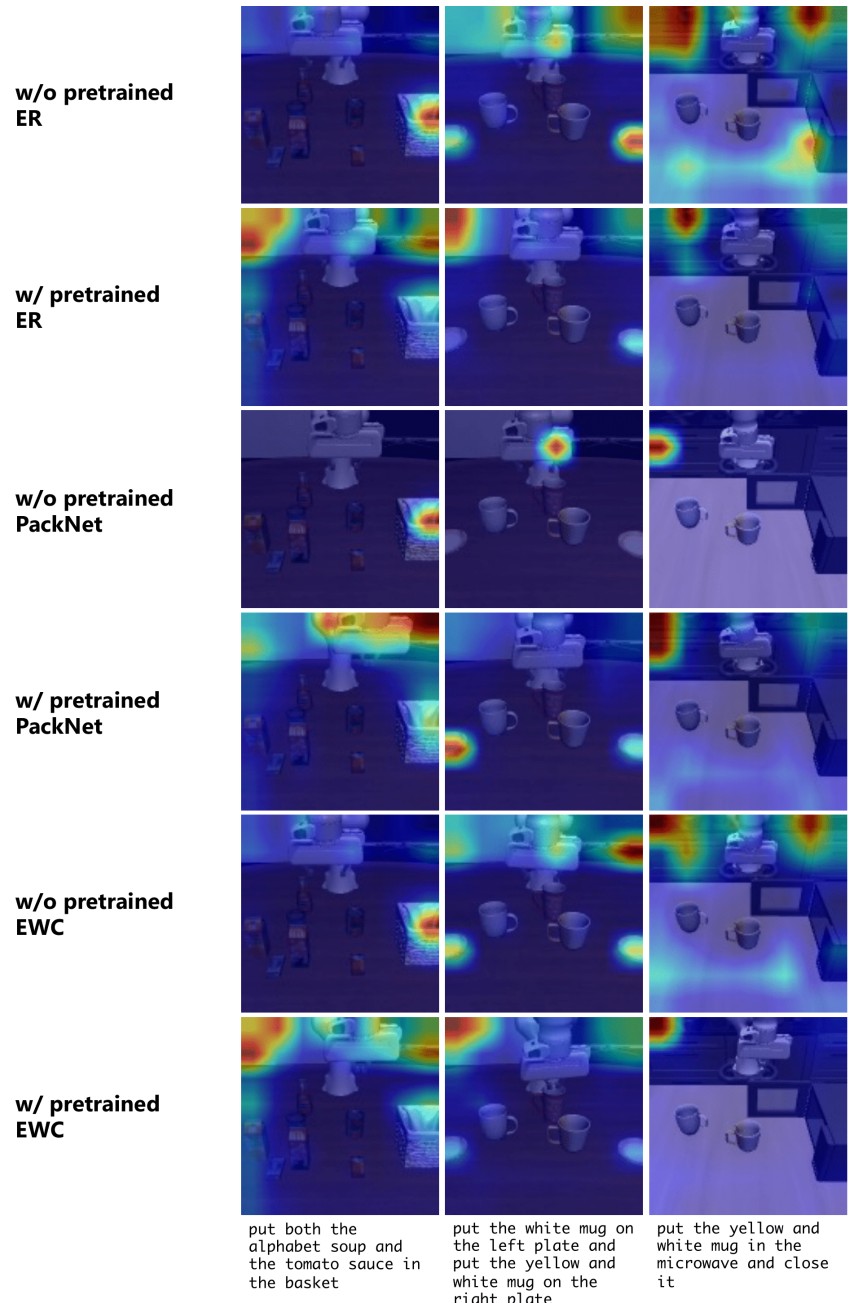

Figure 26: Attention map comparison between models without/with pretrained models using RESNET-T and different lifelong learning algorithms on three selected tasks from LIBERO-LONG.

*Findings:* Figure 26 shows attention map comparisons between models with/without pretrained models on LIBERO-LONG with RESNET-T and all three LL algorithms. We observe that:

1. With pretraining, the policies attend to task-irrelevant regions more easily than those without pretraining.

2. Some of the policies with pretraining have better attention to the task-relevant features than their counterparts without pertaining, but their performance remains lower (the last in the second row and the second in the fourth row). This observation, again, shows that there is no positive correlation between semantically meaningful attention maps and the policy's performance.

