# OpenReview forum: "LIBERO: Benchmarking Knowledge Transfer for Lifelong Robot Learning"
_robot-learning.org/CoRL/2023/Workshop/OOD — OOD Workshop @ CoRL 2023_

### Official Review · Reviewer_5hgM · 2023-10-16
**Great work on benchmarking for lifelong robot learning**

**Rating:** 9
**Confidence:** 4

**Review:**

This work provides a suite of benchmark tasks to systematically study the effects of distribution shifts, neural architecture & algorithmic design, task ordering, and pretraining. The paper provides insights from comparisons between different LLDM algorithms that I enjoyed learning about. This work is very relevant to the workshop focus and addresses the need for benchmarking for robotic applications.

My only suggestion is to add a discussion on how this work is different from existing benchmarks for manipulation applications. The benchmarking workshop from CoRL 2022 discussed a number of such works.

---

### Official Review · Reviewer_aHVK · 2023-10-17
**Good benchmark for lifelong learning**

**Rating:** 7
**Confidence:** 4

**Review:**

The paper presents a new benchmark that will be very useful for the lifelong learning community in robotics. There are some interesting results presented having to do with pretraining, finetuning, and algorithm design, but it would be good to see more results having to do with applying LIBERO to model distribution shifts between tasks over time, for example covariate shifts, shifts following different time scales, etc. In particular since LIBERO uses language conditioning, there is the question of how to model realistic distribution shifts using language.

In terms of generalization using procedural knowledge, there is only a focus on task goals and not the intermediate steps to the goal, which can also be interpreted as part of the procedure. It could be nice to create task suites based on the simpler tasks or primitives involved in more complex tasks.

While some of the research conclusions presented are unexpected, others do not seem as surprising; for example, it seems expected that transformers perform better than RNNs on time-series data. Since lifelong learning optimizes for backward as well as forward transfer, it makes sense that sequential finetuning would be better at forward transfer.

The benchmark presented is well-assembled and from the results given promises to allow for even more results to be produced by others in the future.

---

### Decision · Program_Chairs · 2023-10-17

**Decision:**

Accept

**Comment:**

We agree with the reviewers’ assessment that this work is technically sound and will contribute to productive, topical discussions at the 2023 Workshop on OOD Generalization in Robotics. In particular, we appreciate that this work highlights two fundamental aspects of the OOD challenge; 1) Continual/Lifelong learning is necessary for robots to maintain performance when distributions change over time, and 2) benchmarks specific to OOD challenges in robotics are needed. We recommend the authors incorporate the reviewers’ feedback into their camera-ready submission to further improve their manuscript.